# Strikingly distinctive NH₃-SCR behavior over Cu-SSZ-13 in the presence of NO₂

Yulong Shan ®[1], Guangzhi He ®[1] ✉, Jinpeng Du[2], Yu Sun[1,3], Zhongqi Liu[1,3], Yu Fu[1,3], Fudong Liu ®[4], Xiaoyan Shi[1,3], Yunbo Yu[1,2,3] & Hong He ®[1,2,3] ✉

Commercial Cu-exchanged small-pore SSZ-13 (Cu-SSZ-13) zeolite catalysts are highly active for the standard selective catalytic reduction (SCR) of NO with NH₃. However, their activity is unexpectedly inhibited in the presence of NO₂ at low temperatures. This is strikingly distinct from the NO₂-accelerated $NO_x$ conversion over other typical SCR catalyst systems. Here, we combine kinetic experiments, in situ X-ray absorption spectroscopy, and density functional theory (DFT) calculations to obtain direct evidence that under reaction conditions, strong oxidation by NO₂ forces Cu ions to exist mainly as Cu^II species (fw-Cu²⁺ and NH₃-solvated Cu^II with high CNs), which impedes the mobility of Cu species. The SCR reaction occurring at these Cu^II sites with weak mobility shows a higher energy barrier than that of the standard SCR reaction on dynamic binuclear sites. Moreover, the NO₂-involved SCR reaction tends to occur at the Brønsted acid sites (BASs) rather than the Cu^II sites. This work clearly explains the strikingly distinctive selective catalytic behavior in this zeolite system.

Increasingly stringent mobile source emission regulations have been pursued around the world to tackle environmental pollution. Nitrogen oxides ($NO_x$) are inevitable gaseous pollutants emitted from internal combustion engines. Selective catalytic reduction of $NO_x$ with NH₃ (NH₃-SCR) is the most widely adopted technology for the removal of $NO_x$ from diesel engines[1,2]. The successful commercialization of Cu-SSZ-13 as an NH₃-SCR catalyst is a significant achievement for diesel engine exhaust post treatment[3]. In the past decade, numerous studies have endeavored to uncover the standard SCR (SSCR) reaction mechanism[4–7], hydrothermal deactivation mechanism[8–11], and SO₂ poisoning deactivation mechanism[12–14], and to develop economic and sustainable synthesis methods for Cu-SSZ-13[15–18], bringing about continuous optimization of Cu-SSZ-13 for commercial SCR catalysts.

In actual application, a diesel oxidation catalyst (DOC) is utilized to oxidize carbon monoxide (CO) and hydrocarbons (HCs), accompanied by partial oxidation of NO to NO₂. The formed NO₂ can participate in the NH₃-SCR process through the so-called "fast SCR" reaction (FSCR, reaction 1, consisting of reactions 2 and 3). It is generally believed that the deNO$_x$ efficiency of the FSCR reaction should be higher than that of SSCR (reaction 4) due to bypassing the oxidation of NO, which is usually the rate-limiting step in the SSCR reaction on V-based and Fe-zeolite catalysts[19,20].

$$NO + NO_2 + 2NH_3 \rightarrow 2N_2 + 3H_2O \tag{1}$$

$$2NO_2 + 2NH_3 \rightarrow NH_4NO_3 + N_2 + H_2O \tag{2}$$

$$NO + NH_4NO_3 \rightarrow N_2 + NO_2 + 2H_2O \tag{3}$$

$$4NO + 4NH_3 + O_2 \rightarrow 4N_2 + 6H_2O \tag{4}$$

[1]State Key Joint Laboratory of Environment Simulation and Pollution Control, Research Center for Eco-Environmental Sciences, Chinese Academy of Sciences, Beijing 100085, China. [2]Center for Excellence in Regional Atmospheric Environment and Key Laboratory of Urban Pollutant Conversion, Institute of Urban Environment, Chinese Academy of Sciences, Xiamen 361021, China. [3]University of Chinese Academy of Sciences, Beijing 100049, China. [4]Department of Civil, Environmental, and Construction Engineering, Catalysis Cluster for Renewable Energy and Chemical Transformations (REACT), NanoScience Technology Center (NSTC), University of Central Florida, Orlando, FL 32816, USA. ✉e-mail: gzhhe@rcees.ac.cn; honghe@rcees.ac.cn

However, there have been few studies reporting that $NO_2$ measurably promotes the $NH_3$-SCR efficiency over Cu-SSZ-13 catalytic systems. On the contrary, inhibition of NO conversion by $NO_2$ was found over Al-rich Cu-SSZ-13 catalysts due to $NH_4NO_3$ formation, which is the so-called "abnormal fast $NH_3$-SCR reaction"[21]. In our recent study, we found that the inhibiting effect of $NO_2$ was closely related to Brønsted acid sites (BASs) and can be alleviated by hydrothermal aging due to the decrease in the number of BASs in Cu-SSZ-13[22]. Therefore, we speculated that $NO_2$ reduction probably occurs at BASs. Also, we previously observed the reaction between NO and $NH_4NO_3$ occurring at BASs over the H-SSZ-13 catalyst[23]. Furthermore, Kubota et al. found that NO reacts with $NH_4NO_3$ more rapidly than $NH_4NO_3$ decomposition over H-AFX and H-CHA zeolites[24,25]. However, the situation in Cu-containing zeolites is more complicated. McEwen et al. found that four-fold-coordinated Cu(II) species dominate the Cu-SSZ-13 catalyst under FSCR conditions, which differs from the composition under SSCR conditions, where Cu(I) and Cu(II) species both exist[26]. Paolucci et al. investigated the oxidation process of $Cu(I)(NH_3)_2$ species by $O_2$ and $NO_2$. It was found that oxidation by $NO_2$ occurred at isolated Cu sites, rather than at the Cu dimer sites required for $O_2$ activation[5]. More recently, Liu et al. investigated the FSCR mechanism over the Cu-OH site on Cu-CHA zeolite and showed the important role of BASs in the FSCR reaction[27]. Therefore, it can be concluded that the FSCR reaction pathway over Cu-SSZ-13 is unique and different from other catalytic systems where $NO_2$ accelerates SCR rates. The active sites as well as redox pathways may change over Cu-SSZ-13 in the presence of $NO_2$. Compared to the relatively few studies on the FSCR reaction mechanism, researchers have conducted numerous experimental and theoretical studies to explore the SSCR reaction mechanism in the past decade. Thus, the SSCR mechanism has been relatively clear, in which dynamic binuclear $Cu^+$ species are the primary active sites[4,5,28]. However, the influence of $NO_2$ on the active Cu sites and the mechanism of the $NO_2$-involved SCR reaction are barely discussed, and are worth exploring since NO and $NO_2$ always coexist in actual applications.

In this study, the SCR reaction over the Cu-SSZ-13 catalyst in the presence of both NO and $NO_2$ was studied by kinetic measurements. In situ X-ray absorption fine structure (XAFS) measurements were applied to reveal the state of copper species under SSCR (with only NO as $NO_x$), FSCR (equal mixture of NO and $NO_2$ as $NO_x$) and $NO_2$-SCR (only $NO_2$ as $NO_x$) reaction conditions. Density functional theory (DFT) calculations were conducted to identify the $NO_2$-involved SCR reaction

pathways. These results provide new insights into the role of $NO_2$ in the $NH_3$-SCR reaction and shed light on the actual application of Cu-SSZ-13 catalysts in the presence of both NO and $NO_2$.

## Results and discussion

### Kinetic studies of $NO_x$ conversion under SSCR, FSCR and $NO_2$-SCR conditions

We first carried out kinetic studies on the SSCR reaction, with the results shown in Fig. 1 and Supplementary Fig. 1. The SSCR rate increases linearly with the square of Cu loading when the Cu loading is below 1.7 wt.% (magnified in Fig. 1b), indicating the participation of Cu pairs in the standard $NH_3$-SCR reaction. Previous studies have reported that $Cu^I$ dimers are formed with $O_2$ activation in the oxidation half-cycle ($Cu^I$→$Cu^{II}$)[4,5]. Recently, Hu et al. also proposed a $Cu^{II}$-pair-mediated low-temperature reduction half-cycle ($Cu^{II}$→$Cu^I$)[6]. Chen et al. also indicated the participation of Cu pairs in the reduction half-cycle[29]. Therefore, the formation of a Cu pair in the same cage is significantly important for the overall standard $NH_3$-SCR reaction process. The increase trend slows down with further rise in the Cu loading. The turnover frequency (TOF) shows a volcano-type tendency, with a maximum at Cu loading of 1.7 wt.% (Fig. 1c). The increase in TOF at low Cu loading is attributed to the quadratic increase in the SSCR rate. At high Cu loading, however, the decline of TOF is probably due to the underutilization of the active Cu sites. According to the calculation method reported by Jones et al.[30], every 2.4 and 3.5 CHA cages contain one Cu ion for $Cu_{3.8}$-SSZ-13 and $Cu_{2.6}$-SSZ-13 samples, respectively. The formed $Cu-NH_3$ complex or dimer Cu species under SSCR conditions probably impede the access of reactants to the Cu ions deep inside the pores, causing inefficiency in the use of Cu ions[31]. The activation energy (Ea) and pre-exponential factor (A) both increase with the increase in Cu loading, which was also observed by Gao et al[31]. Recently, Krisha et al. reported that the Ea of $Cu^I$ oxidation increased monotonically with Cu density in a fixed kinetic regime due to the non-mean-field behavior of Cu-SSZ-13 in the $NH_3$-SCR reaction and that the Ea of $Cu^{II}$ reduction was unchanged when the Cu load was higher than 0.69 wt.%[32]. On the other hand, the kinetic relevance of $Cu^{II}$ reduction increased with increasing Cu ion density, the Ea of which was higher than that of $Cu^I$ oxidation[30,32]. Therefore, the increase of the Ea in $Cu^I$ oxidation and kinetic relevance of $Cu^{II}$ reduction both contributed to the increase in the Ea of the SSCR reaction.

Then, the FSCR reaction over Cu-SSZ-13 was carried out as shown in Supplementary Figs. 2a and 3a. Compared to the SSCR reaction, the

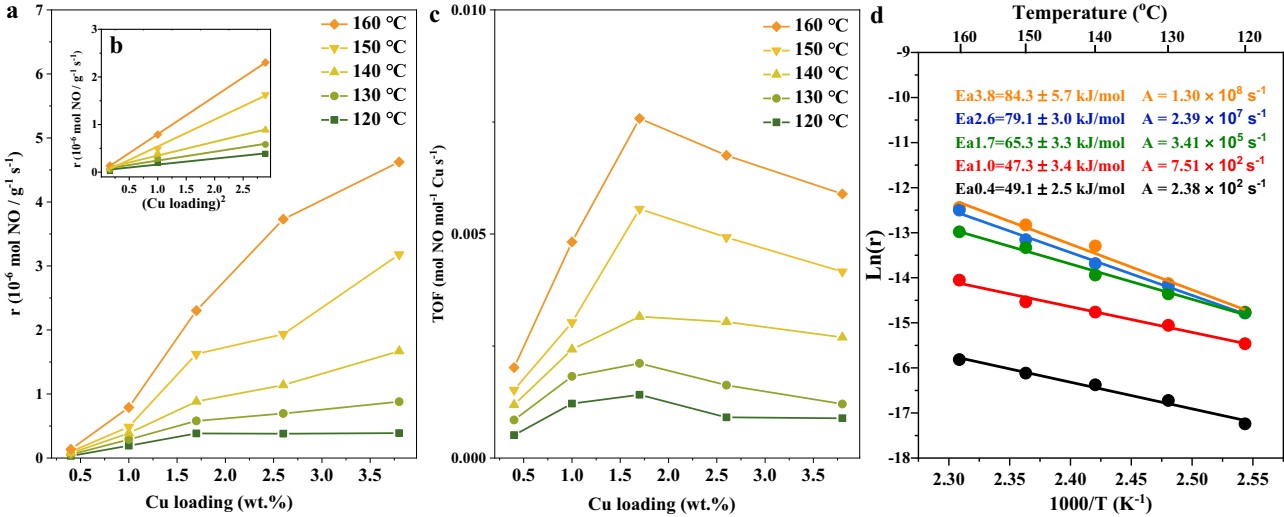

**Fig. 1 | Kinetic analysis of standard SCR reaction. a** SSCR reaction rates as a function of Cu loading. **b** SSCR reaction rates as a function of the square of Cu loading. **c** SSCR turnover frequencies (TOF) as a function of Cu loading. **d** Activation energies (Ea) and pre-exponential factors (A) with different Cu loadings.

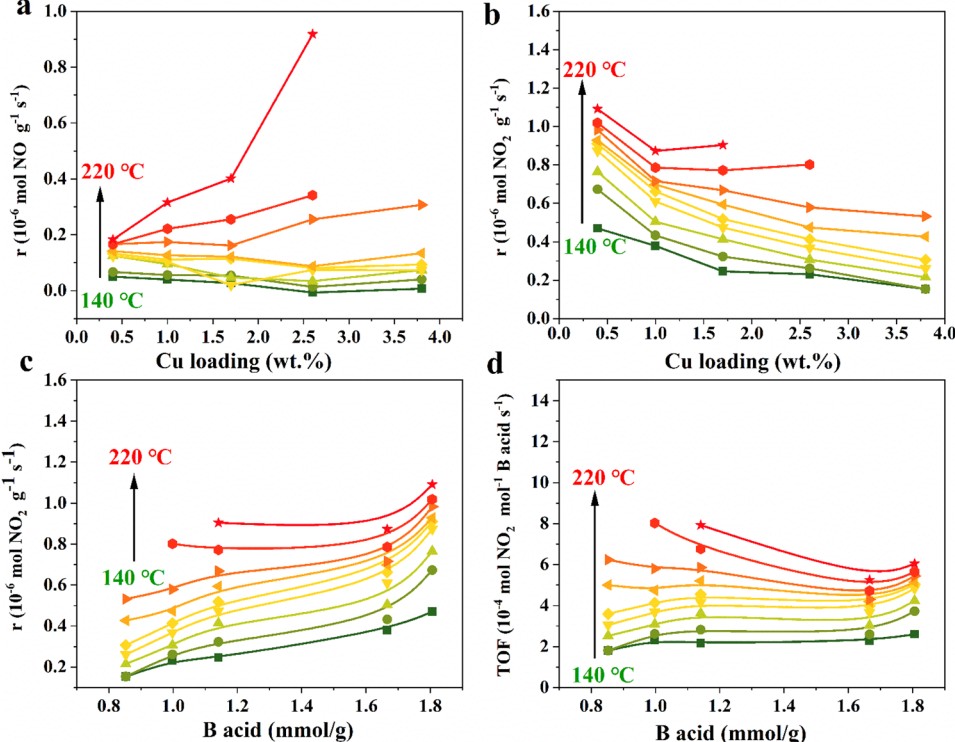

**Fig. 2 | Kinetic analysis of fast SCR reaction. a** NO reaction rates as a function of Cu loading over Cu-SSZ-13 catalysts in NO and NO$_2$ gas mixtures. **b** NO$_2$ reaction rates as a function of Cu loading over Cu-SSZ-13 in NO and NO$_2$ gas mixtures. **c** NO$_2$ reaction rates as a function of BASs over Cu-SSZ-13 in NO and NO$_2$ gas mixtures. **d** NO$_2$ turnover frequencies (TOFs) as a function of BASs in NO and NO$_2$ gas mixtures.

NO$_x$ conversion over Cu3.8-SSZ-13 was significantly inhibited in the presence of NO$_2$, which was strikingly distinct from the NO$_2$-accelerated NO$_x$ conversion over Fe-based zeolite and oxide catalysts (Supplementary Fig. 3). Supplementary Fig. 2 shows the NO$_x$, NO and NO$_2$ conversion levels over Cu-SSZ-13 with different Cu loadings under steady-state FSCR conditions. We normalized the NO and NO$_2$ reaction rates by the catalyst weight as a function of Cu loading, with the results shown in Fig. 2a, b, respectively. The NO consumption rates under FSCR and SSCR condition were compared (Supplementary Fig. 4) and the result showed that NO reduction was severely suppressed at low temperatures under FSCR conditions. The extremely low NO conversion at low temperatures was previously thought to be resulted from zeolite pore blocking by the formation of stable NH$_4$NO$_3$[21,23]. The NH$_4$NO$_3$ formation was verified by the observation of N$_2$O in an FSCR-TPD experiment (Supplementary Fig. 5), since the N$_2$O mainly originated from NH$_4$NO$_3$ decomposition. Interestingly, the NO$_2$ reduction markedly decreased with the increase in Cu loading, while it increased as the number of BASs rose at low temperatures (Fig. 2b, c and Supplementary Fig. 6). This demonstrated that the block of active sites by NH$_4$NO$_3$ was not the only reason for the NO$_2$-inhibition effects, otherwise both NO and NO$_2$ reduction were inhibited. The BASs primarily participated in the reduction of NO$_2$, which was also observed in the NO$_2$-SCR reaction (Supplementary Fig. 7). Moreover, the turnover frequency (TOF) of NO$_2$ on BASs hardly changed as the number of BAS varied. Supplementary Fig. 8 presents the NO$_2$ reaction rate as a function of Cu loading and BASs under NO$_2$-SCR conditions, which showed the same trend as that in the co-existence of NO and NO$_2$. Moreover, we carried out the NO$_2$-SCR reaction over H-SSZ-13 and Cu$_{2.6}$-SSZ-13 with different Si/Al ratios and found that the zeolites with low Si/Al exhibited high NO$_x$ conversion due to their high numbers of BASs at low temperatures (Supplementary Fig. 9). The above results indicated that NO$_2$ primarily reacted at BASs while NO was difficult to be reduced in the presence of NO$_2$. NO$_2$ disproportionation occurs

on the BASs to form nitrates and adsorbed NO$^+$, which then react with NH$_3$ to form NH$_4$NO$_3$ and NH$_2$NO, respectively[33–35]. It is generally known that NO can be effectively reduced at Cu sites. However, the formation of NH$_4$NO$_3$ impedes NO access to the active Cu sites. Instead, NO reacts with NH$_4$NO$_3$ at BASs to form N$_2$ and NO$_2$ through reaction (3) (TPSR shown in Supplementary Fig. 10). Furthermore, the NO and NO$_2$ conversion levels over Cu2.6-SSZ-13 and Cu0.4-SSZ-13 under SSCR, FSCR and NO$_2$-SCR conditions are separately depicted in Supplementary Fig. 11. For Cu2.6-SSZ-13 sample, the NO conversion under SSCR conditions was remarkably higher than that under FSCR conditions, which indicated that the SSCR reaction pathway was significantly inhibited under FSCR conditions. We ascribed the low NO conversion to the reaction with NH$_4$NO$_3$ (i.e., FSCR reaction) and the extra NO$_2$ conversion to the reaction between NO$_2$ and NH$_3$. For Cu0.4-SSZ-13 sample, the NO conversion under FSCR conditions was likewise inhibited compared to that under SSCR conditions. Differently, the NO$_2$ conversion under FSCR and NO$_2$-SCR conditions were relatively higher than the NO conversion under SSCR conditions due to the insufficient Cu active sites for SSCR reaction. As a result, the FSCR rates of NO$_x$ can also be higher than SSCR rates of NO$_x$ especially when the Cu-zeolite behaves low NO conversion (low Cu loadings, hydrothermal aging state, etc.), which was observed in previous studies[23,27,36,37]. In another word, the NO conversion was inhibited in the presence of NO$_2$, while the effect of NO$_2$ on NO$_x$ conversion was uncertain and relates to NO$_2$ conversion under FSCR conditions as well as NO$_x$ conversion under SSCR conditions.

**Wavelet transform analysis of in situ EXAFS measurements**
Further, we conducted in situ XAFS measurements on Cu-SSZ-13 samples to uncover the valence state and coordination of copper species under different conditions. Wavelet transform (WT) analysis of extended X-ray absorption fine structure (EXAFS) spectra is a powerful technique to resolve overlapping contributions from different

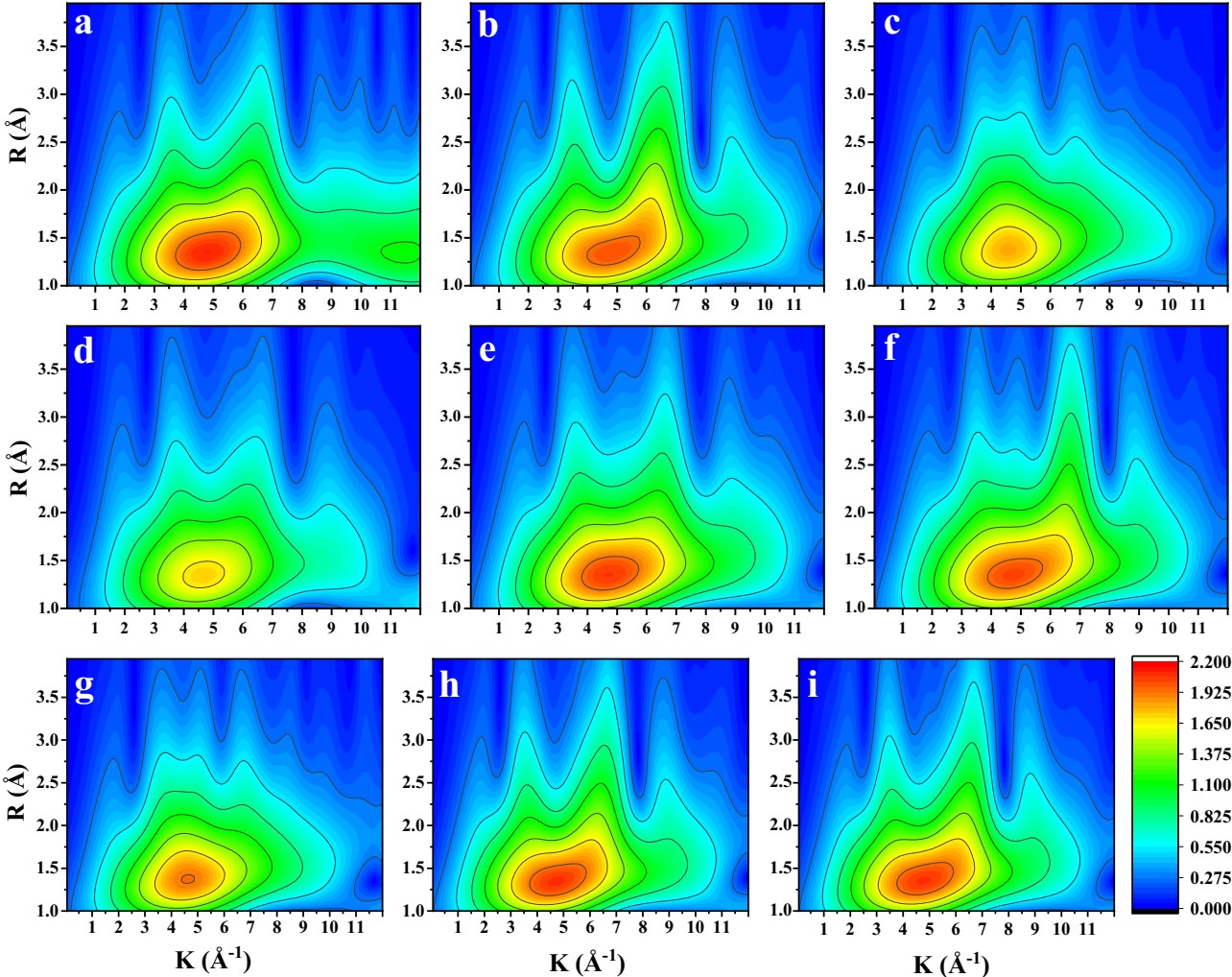

**Fig. 3 | WT plots for EXAFS spectra of Cu-SSZ-13 treated under different conditions at 200 °C. a** Cu-SSZ-13 pretreated in $O_2$/He. **b** NO adsorption. **c** $NH_3$ adsorption. **d** NO + $NH_3$ adsorption. **e** NO + $NH_3$ co-adsorption followed by reaction with $O_2$. **f** NO + $NH_3$ co-adsorption followed by reaction with $NO_2$. **g** SSCR conditions. **h** FSCR conditions. **i** $NO_2$-SCR conditions.

neighbor atoms at close distances around the absorber. As shown in Fig. 3a, the pretreated sample shows a distinct first shell peak at (4.5 Å⁻¹, 1.3 Å), which is associated with contributions from framework oxygen atoms. This result suggested that the copper species mainly exist as fw-$Cu^{2+}$ species, which have high coordination numbers[28]. For the second shell sphere (R(Å) > 2 Å), two lobes, at (3.5 Å⁻¹,2.8 Å) and (6.5 Å⁻¹ 3.3 Å), are well-resolved due to the different backscattering properties of various atoms, which strongly depend on the atomic number. The first lobe is assigned to the second-shell oxygen atom due to the low k value of oxygen atoms. The latter one is attributed to the signals from the Si or Al atoms of the framework. Although some studies attributed the latter lobe to the Cu-Cu contributions in oxygen-bridged Cu dimers[38,39], we scarcely observed $CuO_x$ species in X-ray absorption near edge structure (XANES) and EXAFS profiles (Supplementary Fig. 12) and did not carry out the procedure of introducing $O_2$ to $NH_3$-treated Cu-SSZ-13 to form oxygen-bridged Cu dimers with four $NH_3$ ligands. Therefore, we deduced that the lobe at 6.5 Å⁻¹ is primarily derived from the framework Si or Al atoms in the second shell in this work. In fact, the copper species in Cu-SSZ-13 are initially in the solvated state as $[Cu(H_2O)n]^{2+}$ under ambient conditions, which weakens the interaction between copper species and the zeolite framework[28,40]. High-temperature treatment in $O_2$/$N_2$ removes the coordinated water molecules and oxidizes copper species to $Cu^{2+}$. As a result, the copper

species are in a high valence state and strongly interact with the zeolite framework through electrostatic forces.

After NO adsorption, $Cu^{2+}$ ions are partially reduced, resulting in a slight decrease in the coordination numbers (CNs) of the first shell, denoted by the decrease and weakening of the colored area (Fig. 3b). The lobes resulting from the contributions of the second shell stretched to (3.5 Å⁻¹, 3.1 Å) and (6.5 Å⁻¹, 3.7 Å), respectively. When the pretreated sample was exposed to an $NH_3$ or NO + $NH_3$ atmosphere, the signal of the first shell sharply decreased (Fig. 3c, d), suggesting that the CNs of the Cu ions significantly declined due to their reduction. Moreover, the two lobes are not well-resolved in the spectra, indicating a decrease in the scattering from the second shell. This is consistent with the formation of dynamic $[Cu(NH_3)_2]^+$ species, which is supported by the appearance of feature B in Supplementary Fig. 13a after $NH_3$ or NO + $NH_3$ adsorption. After oxidation by $O_2$ and $NO_2$, the CNs of the first shell increased to a level similar to that of the pretreated sample, accompanied by the formation of two well-resolved lobes at the second shell (Fig. 3e, f). This demonstrated that $Cu(NH_3)_2^+$ species are oxidized into $Cu^{II}$ ions and that the interaction between the $Cu^{2+}$ ions and the zeolite framework is recovered. Besides the scattering by framework Si (or Al), the second lobe at 6.5 Å⁻¹ probably resulted from the scattering of the second shell Cu species, since oxygen-bridged Cu dimers are formed after $Cu^I(NH_3)_2$ oxidation by $O_2$[5,38]. Compared with

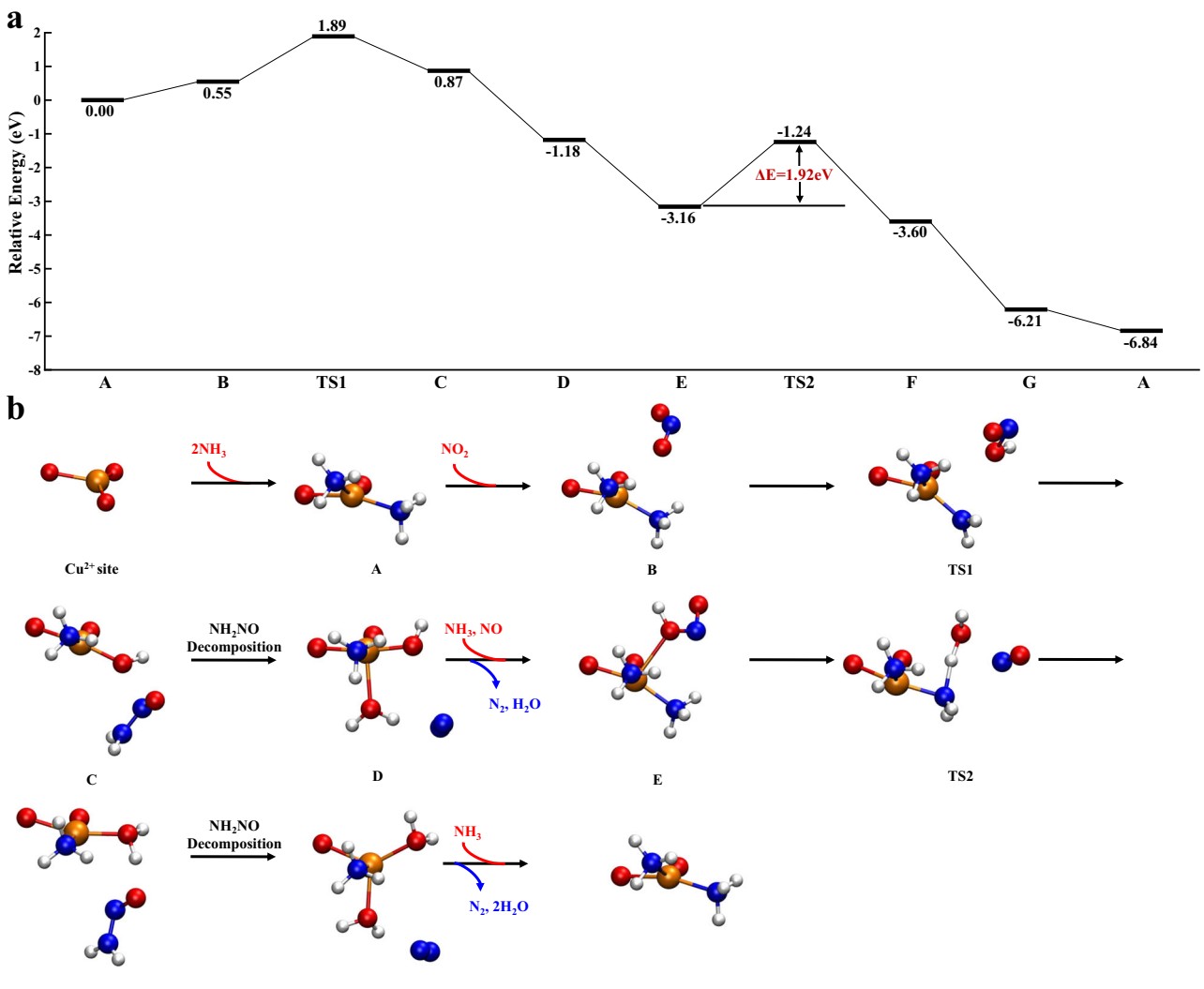

**Fig. 4 | Reaction pathway of the fast SCR cycle at Z$_2$Cu$^{II}$ site. a** Gibbs free energy profile. **b** Optimized geometries of the reactants, transition states (TSs) and products for all elementary steps are presented in the lower panel. Except for the O atoms linked to the Cu$^{2+}$ ion, all other atoms of the zeolite framework are omitted for clarity. Orange, red, blue and white circles denote Cu, O, N and H atoms, respectively.

oxidation by O$_2$, oxidation by NO$_2$ resulted in a higher signal for the lobe at -6.5 Å$^{-1}$, indicating that more Cu$^I$ species are oxidized into Cu$^{II}$ ions (fw-Cu$^{II}$ or NH$_3$-solvated Cu$^{II}$ species with high CNs) during the reaction with NO$_2$. This phenomenon is consistent with the result reported by Paolucci et al. showing that NO$_2$ can oxidize the residual Cu$^I$(NH$_3$)$_2$ species that cannot be oxidized by O$_2$. As also reported by Paolucci, the transient oxidation of Cu$^I$(NH$_3$)$_2$ species by NO$_2$ is a single-site process without formation of Cu dimers. Therefore, it can be inferred that the presence of NO$_2$ probably changed the SCR reaction active sites from dimer Cu to isolated Cu species, which further influence the SSCR reaction. This deduction indicated that most Cu species are bonded with the zeolite framework and that the mobility of Cu species is limited during the process of Cu$^I$(NH$_3$)$_2$ oxidation by NO$_2$. Although the transient reaction can reflect the Cu state and coordination during half-cycles, it was deemed more meaningful to identify the Cu species under FSCR reaction conditions.

Figures 4g–i depicts 2D plots of the WT EXAFS spectra under SSCR, FSCR and NO$_2$-SCR conditions. Under SSCR conditions, the WT EXAFS spectra resemble the ones in Fig. 3c, d. The first shell peak weakened under SSCR conditions, indicating a decrease in the CNs of Cu species. The absence of the lobes at the second shell suggests the easy mobility of the copper complex due to the NH$_3$

solvation effect. In the presence of NO$_2$, however, the CNs of the first shell significantly increased, indicating the oxidation of copper species, which was also supported by the results of McEwen et al.[26]. Moreover, two well-resolved lobes at the second shell are observed, suggesting that oxidization leads to the copper species becoming closely coordinated with the zeolite framework, which limits their mobility during the SCR reaction. The WT EXAFS spectra are consistent with the Fourier-transformed (FT) EXAFS results (Supplementary Fig. 14 and Table 1), which are discussed in detail in the Supporting Information. The above results proved the existence of greater amounts of dynamic Cu$^I$(NH$_3$)$_2$ species under SSCR reaction conditions than that under FSCR and NO$_2$-SCR reaction conditions. Notably, although we proved the existence of significant framework-bound Cu$^{II}$ species under FSCR conditions, the NH$_3$-solvated Cu$^{II}$ species cannot be ruled out by the XAFS experiment. Indeed, the NH$_3$-solvated Cu$^{II}$ species existed, as indicated by the observation of NH$_3$ desorption from Cu sites in FSCR-TPD profiles (Supplementary Fig. 5), which was consistent with the computed phase diagram reported by Paolucci et al.[28]. Therefore, we next turned to the DFT calculation to investigate the possible FSCR reaction pathways over fw- and NH$_3$-solvated Cu$^{II}$ [Cu$^{2+}$ and (Cu$^{II}$OH)$^+$] species, BASs and dimer Cu species.

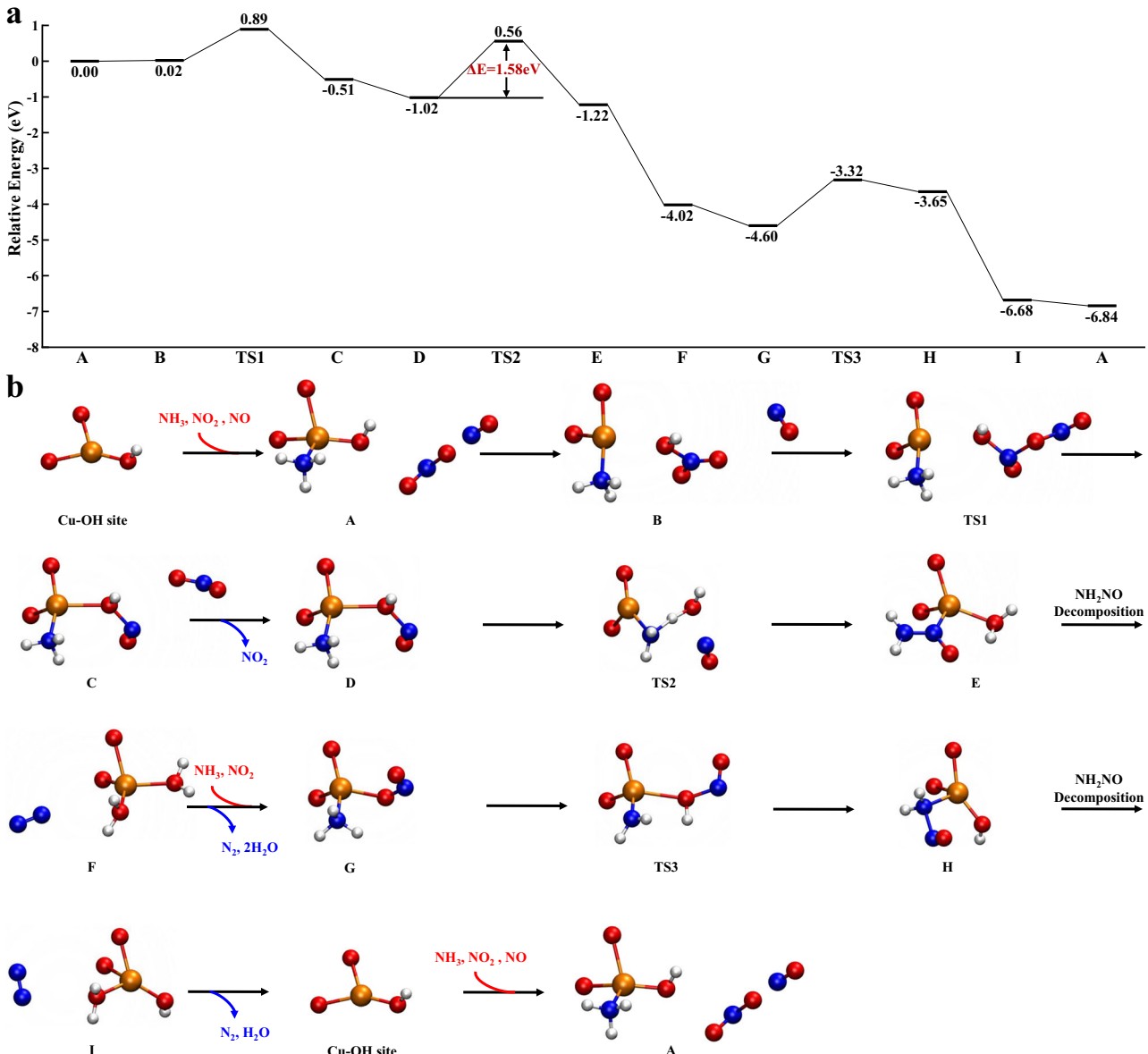

**Fig. 5 | Reaction pathway of the fast SCR cycle at ZCuᴵᴵOH site. a** Gibbs free energy profile. **b** Optimized geometries of the reactants, TSs and products for all elementary steps are presented in the lower panel. Except for the two O atoms linked to the Cu-OH group, all other atoms of the zeolite framework are omitted for clarity. All legends are the same as those in Fig. 4.

## DFT calculation

We first calculated the FSCR reaction pathway over fw-Cuᴵᴵ species (Fig. 4). The framework-bound Cuᴵᴵ first adsorbs two $NH_3$ molecules without separation from the framework, which then interacts with $NO_2$ to form $Z_2Cu^{II}NH_3OH$ and $NH_2NO$ species (B → C). The $Z_2Cu^{II}NH_3OH$ further adsorbs an $NH_3$ molecule and reacts with NO, resulting in the formation of $Z_2Cu^{II}NH_3$, $NH_2NO$ and $H_2O$ (E → F), which was predicted to be the rate-determining step of the SCR reaction cycle with a high energy barrier of 1.92 eV. The formed $NH_2NO$ is easily decomposed into $N_2$ and $H_2O$ through a series of H-migration and isomerization processes (Supplementary Fig. 16)[29]. Last, the gaseous $NH_3$ molecules are supplied to regenerate the initial A species.

Next, the FSCR reaction pathway over ZCuᴵᴵOH was calculated and depicted in Fig. 5. ZCuᴵᴵOH first adsorbs an $NH_3$ molecule to reach a coordinatively saturated state, which interacts with $NO_2$ to form an $HNO_3$ molecule without any energy barrier. The B species is actually considered to be $NH_4NO_3$ adsorbed on Cu sites. Next, the adsorbed $HNO_3$ reacts with NO from the gas phase with an energy barrier of

0.87 eV, resulting in the formation of adsorbed $HNO_2$ and the release of an $NO_2$ molecule (C → D). Then, the adsorbed $HNO_2$ reacts with the $NH_3$ ligand to generate $NH_2NO$ and $H_2O$. As the desorption of $N_2$ and $H_2O$ molecules occurs, $NH_3$ and $NO_2$ are adsorbed at the Cu site and react to generate $NH_2NO$ and -OH groups. With the decomposition of $NH_2NO$ into $N_2$ and $H_2O$, the ZCuᴵᴵOH site is regenerated. The rate-determining step of the FSCR cycle over the ZCuᴵᴵOH site corresponds to the reaction of adsorbed $HNO_2$ with an $NH_3$ ligand to produce $NH_2NO$ and $H_2O$ (E → F), with an energy barrier of 1.58 eV.

In addition, the FSCR reaction pathways over $NH_3$-solvated Cuᴵᴵ species [Cuᴵᴵ and (CuᴵᴵOH)⁺] were also calculated and presented in Supplementary Fig. 17, 18. All the energy barriers were found to be relatively high (1.54 and 1.65 eV). Moreover, we consider the possibility that various $NH_3$-solvated Cuᴵᴵ species diffuse into an adjacent cage to form Cuᴵᴵ-pairs as shown in Supplementary Fig. S15. As expected, the formation of Cuᴵᴵ pairs from $Cu^{II}(NH_3)_4$ and $Cu^{II}NO_2(NH_3)_3$ is both thermodynamically and kinetically inhibited due to the steric effect as well as strong interaction with zeolite framework. However, it should

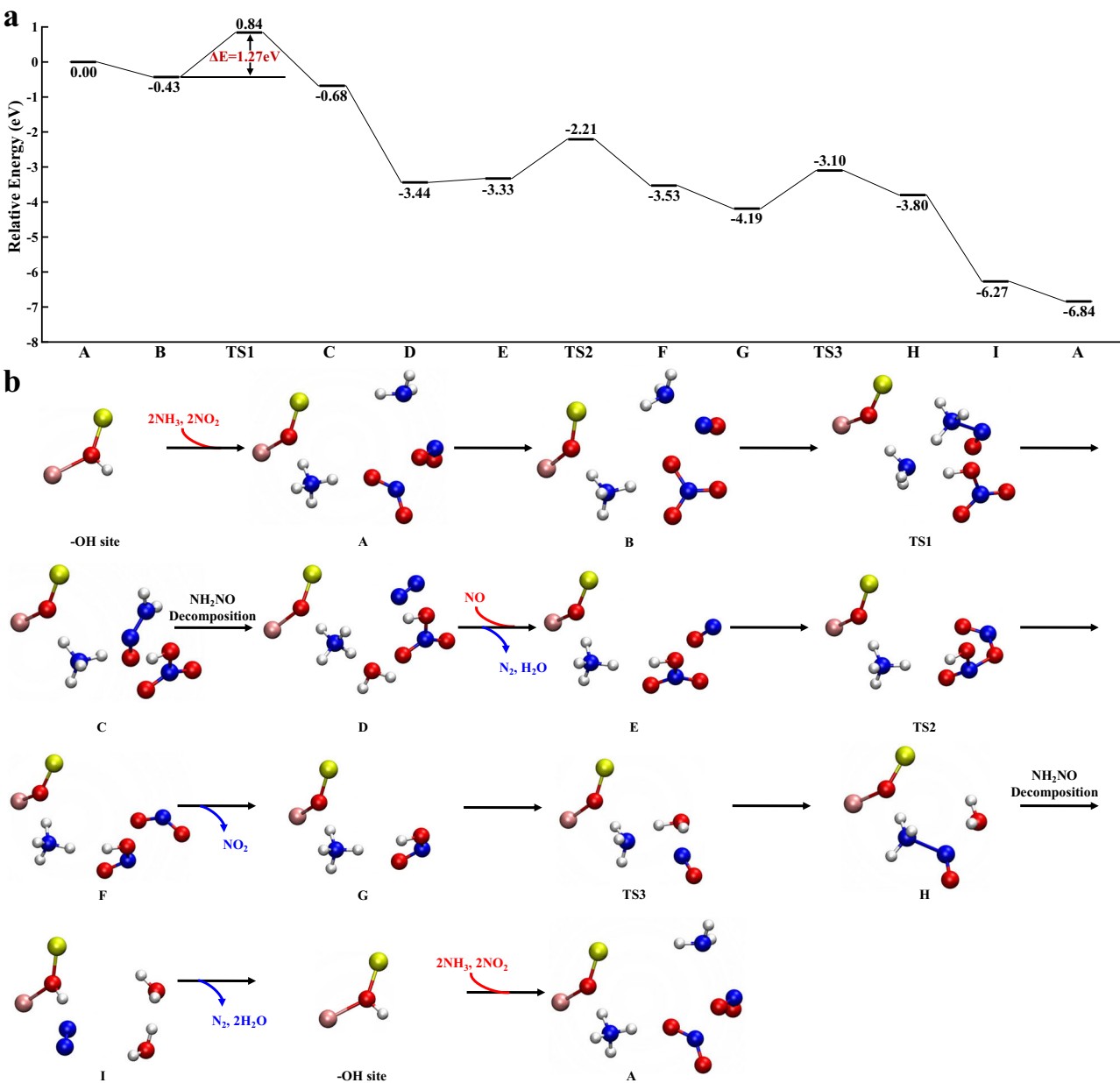

**Fig. 6 | Reaction pathway of the fast SCR cycle at BAS. a** Gibbs free energy profile. **b** Optimized geometries of the reactants, TSs and products for all elementary steps. Except for the Si and Al atoms linked to the OH group, all other atoms of the zeolite framework are omitted for clarity. Yellow and pink circles denote Si and Al atoms, respectively. All other legends are the same as those in Fig. 4.

be noted that $Cu^{II}OH(NH_3)_3$ is different from the other forms since binuclear $Cu^{II}OH(NH_3)_3$ in one cage is thermodynamically more stable than the isolated configuration. Villamaina et al. validated the formation of $Cu^{II}(OH)(NH_3)_x$ dimeric complexes in the oxidation atmosphere through $CO + NH_3$ titration experiment[41]. Hu et al. proposed that $Cu^{II}(OH)(NH_3)$, which is structurally similar to $Cu^I(NH_3)_2^+$ that has one charge and two ligands, acts as inter-cage transportation medium[6]. The Z2Cu$^{II}$ species can transform ZCu$^{II}$(OH) by $NH_3$-assisted hydrolysis to achieve the Cu pairing[42]. However, the regeneration of Cu$^{II}$(OH) in dimeric form showed a high energy barrier of 1.58 eV (A → B in Supplementary Fig. S19), suggesting that the dimeric Cu$^{II}$(OH) species were not highly active in the FSCR reaction.

The FSCR reaction pathway at BASs is displayed in Fig. 6. $NH_3$ is adsorbed on the BASs to form $NH_4^+$ species. Two $NO_2$ molecules interact with the $NH_4^+$ species to form $NH_4NO_3$ species and release an NO molecule without any energy barrier (A → B). The release of NO was also observed in our previous studies during $NO_2$ adsorption on H-SSZ-13 zeolite[43]. Then, NO interacts with an $NH_3$ from the gas phase to form an $NH_3 \cdots NO$ complex, which further reacts with $NH_4NO_3$ to form $NH_4^+$, $HNO_3$, and $NH_2NO$ via an H-migration process (B → C). $NH_2NO$ is decomposed into $N_2$ and $H_2O$. Subsequently, $HNO_3$ reacts with NO from the gas phase, resulting in the formation of $HNO_2$ and the release of an $NO_2$ molecule (E → F). Reaction between $HNO_2$ and $NH_4^+$ species leads to the formation of an $NH_3 \cdots NO$ complex and an $H_2O$ molecule. The $NH_3 \cdots NO$ complex transfers an H atom to regenerate the BAS and changes into $NH_2NO$, which then decomposes into $N_2$ and $H_2O$. The whole catalytic cycle is completed. The overall energy barrier of the FSCR over the Brønsted acid site is 1.27 eV, which corresponds to the reaction between $NH_4NO_3$ and the $NH_3 \cdots NO$ complex, much lower than that over various Cu sites. The DFT-calculated results indicate that the FSCR process in the SSZ-13 zeolite system tends to occur at BASs.

In summary, by combining the analysis of in situ spectroscopic measurements with DFT calculations, we found that $NO_2$ leads to the deep oxidation of copper species as $Cu^{II}$ species (fw-$Cu^{II}$ and $NH_3$-solvated $Cu^{II}$ with high CNs), which significantly inhibits the mobility of Cu sites. As a result, the FSCR reaction occurs primarily at the BASs even though it has a higher energy barrier (1.27 eV) than the locally homogeneous SSCR reaction at dynamic sites (about 1.0 eV). This work reveals the origin of the abnormal $NH_3$-SCR behavior over the commercial Cu-SSZ-13 catalyst in the presence of $NO_2$.

## Methods

### Sample preparation

The initial Cu-SSZ-13 zeolite was in situ synthesized by a one-pot method[15]. The ratio of $Na_2O/Al_2O_3/H_2O/SiO_2/Cu$-TEPA was 3.5/1.0/200/25/3 and the crystallization of the zeolite was performed at 120 °C for 5 days. Due to the excess Cu in the initial product, aftertreatments were required to optimize the Cu contents and distribution. In detail, the as-synthesized Cu-SSZ-13 was post-treated with 0.1 mol/L $HNO_3$ at 80 °C for 12 h to remove $CuO_x$ species. After calcination at 600 °C, the sample was stirred in $NH_4NO_3$ solution (0.01-0.2 mol/L) at 40 °C for the second post-treatment process, followed by filtration, washing, drying and calcination at 600 °C. The obtained Cu-SSZ-13 catalysts were Al-rich zeolites with Si/Al of ~5 and various Cu loadings from 0.4 to 3.8 wt.% (Supplementary Table 2).

### Catalyst evaluation

The standard SCR (SSCR), fast SCR (FSCR) and slow SCR ($NO_2$-SCR) reactions were carried out in a fixed-bed flow reactor system with an online Nicolet Is50 spectrometer, which was used to detect the concentrations of reactants and products. The SSCR conditions included 500 ppm NO and 500 ppm $NH_3$; the FSCR conditions included 250 ppm NO, 250 ppm $NO_2$ and 500 ppm $NH_3$; the $NO_2$-SCR conditions included 300 ppm $NO_2$ and 500 ppm $NH_3$. All the conditions included 3.5% $H_2O$, 5%$O_2$ and $N_2$ balance. The total flow rate was 500 mL/min. The $NO_x$ (NO and $NO_2$) conversion was calculated at steady state:

$$NO\,conversion = \left(1 - \frac{[NO]_{out}}{[NO]_{in}}\right) \times 100\% \quad (5)$$

$$NO_2\,conversion = \left(1 - \frac{[NO_2]_{out}}{[NO_2]_{in}}\right) \times 100\% \quad (6)$$

$$NO_x\,conversion = \left(1 - \frac{[NO]_{out} + [NO_2]_{out}}{[NO]_{in} + [NO_2]_{in}}\right) \times 100\% \quad (7)$$

To conduct the kinetic studies, the gas hourly space velocity (GHSV) was controlled by adjusting the catalyst weight. The GHSV of SSCR, FSCR and $NO_2$-SCR were about ~800,000 $h^{-1}$, ~1,000,0000 $h^{-1}$ and ~2,000,000 $h^{-1}$, respectively. The reaction rates (r) in this study were normalized by catalyst weight based on Eq. (8). The activation energies (Ea) were calculated by the Arrhenius Eq. (9).

$$r = \frac{F_{NO_x} \bullet X_{NO_x}}{W_{cat}} \quad (8)$$

$$r = [NO_x]_0 A e^{\left(-\frac{Ea}{RT}\right)} \quad (9)$$

where $F_{NOx}$ represents the $NO_x$ flow rate (mol/s), $X_{NOx}$ represents the $NO_x$ conversion, $W_{cat}$ is the mass of the catalyst (g), and $[NO_x]_0$ is the inlet concentration of $NO_x$. NOx represents NO, $NO_2$ or a mixture of both.

### Characterization

The elemental composition of the catalysts was measured by inductively coupled plasma atomic emission spectroscopy (ICP-AES). $N_2$ adsorption-desorption analysis of the samples was conducted on a Micromeritics ASAP 2020 instrument. The acid site distribution and contents were measured by $NH_3$ temperature-programmed desorption ($NH_3$-TPD) using the $NH_3$-SCR activity measurement instrument described above. Samples of about 30 mg were used and pretreated in 10% $O_2/N_2$ at 500 °C for 30 min before cooling down to 120 °C. Then, the gas was changed to 500 ppm $NH_3/N_2$ for 60 min, followed by $N_2$ purging for 60 min. Finally, the temperature was raised to 700 °C at a rate of 10 °C/min.

The in situ X-ray absorption fine structure (in situ XAFS) experiments were performed on the 1W1B beamline of Beijing Synchrotron Radiation Facility (BSRF). The absorption data from −200 eV to 800 eV of the Cu K-edge (8979 eV) were collected. The sample was first pretreated in $O_2$/He at 500 °C for 30 min before decreasing the temperature to 200 °C, after which the Pre. spectra were collected. Then, the sample was exposed to 500 ppm $NH_3$/He, 500 ppm NO/He and 500 ppm $NH_3$/He + 500 ppm NO/He for 60 min, respectively, and spectra were collected. After reduction by (NO+$NH_3$)/He, the sample was exposed to 5% $O_2/N_2$ and 500 ppm $NO_2/N_2$ for 60 min, respectively, to obtain the absorption data for the oxidized sample. Moreover, the in situ absorption data were collected after the pretreated samples were exposed to SSCR, FSCR and $NO_2$-SCR atmospheres for 60 min. The X-ray absorption near-edge structure (XANES) data were background-corrected and normalized using the Athena module implemented in the IFFEFIT software package[44]. Extended X-ray absorption fine structure (EXAFS) data were analyzed and fitted using Athena and Artemis (3.0 < k < 13.0 $Å^{-1}$). An amplitude reduction factor ($S_0^2$) of 0.85 was used for all the fitted data sets. Wavelet transform (WT) analysis of the EXAFS was performed to precisely investigate the local coordination environment of copper species.

### Computational details

Spin-polarized periodic DFT calculations were carried out with the Vienna ab initio simulation package (VASP)[45] The Perdew−Burke−Ernzerhof (PBE) generalized gradient approximation was adopted with the van der Waals correction proposed by Grimme (i.e., DFT-D3 method)[46]. The Kohn-Sham orbitals were expanded with a plane-wave basis set with a cutoff energy of 500 eV, and the plane augmented wave (PAW) method was used to describe the interaction between the valence electrons and the cores[47]. The DFT + U method was applied to Cu 3d states with $U_{eff}$ = 6.0 eV to describe the on-site Coulomb interactions[29,48]. During geometrical optimization, the self-consistent-field electronic energies were converged to $1 \times 10^{-5}$ eV and all other atoms were fully relaxed until the maximum force on the atoms was less than $2 \times 10^{-2}$ eV/Å. The Brillouin zone was sampled with a Monkhorst-Pack k-point grid of $1 \times 2 \times 2$. The Gaussian smearing method was utilized, with a smearing width of 0.2 eV. The transition states of elementary steps were located using the climbing image nudged elastic band (CI-NEB) method with several intermediate images between initial and final states[49,50]. Thermodynamic data were processed with the VASPKIT code[51] and the Gibbs free energies were calculated at 200 °C. The SSZ-13 zeolite structure was modelled using two rhombohedral unit cells (24 tetrahedrally coordinated atoms) with size of 18.84 Å × 9.42 Å × 9.42 Å (Supplementary Fig. 20). One Si atom was replaced by one Al atom in each double 6-membered ring, resulting in a model with a Si/Al ratio of 11. One H atom was introduced onto one of the O atoms connected with each Al atom to keep the structure charge-neutral. Based on previous studies[4,29], the present computational settings and models were reliable for investigating the $NH_3$-SCR mechanism over Cu-SSZ-13 zeolites.

## Data availability

All data generated and analyzed in this study are provided in the Article and Supplementary Information, and are also available from the corresponding authors upon reasonable request.

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

## Acknowledgements

This work was supported by the National Natural Science Foundation of China (22188102, 21906172), the Youth Innovation Promotion Associa-tion, CAS (2019045), and the Ozone Formation Mechanism and Control Strategies Project of Research Center for Eco-Environmental Sciences, CAS (RCEES-CYZX-2020). We thank the 1W1B beamline of Beijing Syn-chrotron Radiation Facility for providing the beam time, and thank Lirong Zheng at the Institute of High Energy Physics Chinese Academy of Sciences and Bin Wang at the Sinopec Beijing Research Institute of Chemical Industry for their help in the in situ XAFS experiments.

## Author contributions

H.H. and G.H. designed and supervised the research. Y.L.S. designed and performed the experiments with J.D., Y.S. and Z.L. G.H. and Y.F. conducted the DFT calculations. F.L., X.S, and Y.Y provided suggestions on the manuscript. Y.L.S., G.H. and H.H. wrote the manuscript. All authors discussed the results and commented on the manuscript.

## Competing interests

The authors declare no competing interests.
