## [Peer Review File · Nature Communications]

Title: Strikingly Distinctive NH₃-SCR Behavior over Cu-SSZ-13 in the Presence of NO₂Reviewers' comments:

Reviewer #1 (Remarks to the Author):

Several questions need to be addressed in revision:

1) What is the authors' evidence for NO oxidation being the rate determining step in the SSCR reaction at low T over Cu-SSZ-13? This is well known to be the case on V-based catalysts and on Fe-zeolites, but XAS studies show that the redox state of Cu-SSZ-13 is significantly high at SSCR conditions, which makes the Cu²⁺ reduction a kinetically relevant step.

2) Can the authors rule out the participation of Cu²⁺ dimers in the Cu reduction mechanism of SSCR, as proposed in recent SCR literature (ACS Catal. 10 (2020) 5646; Angew. Chemie 60 (2021) 7197)? Notably, the SSCR rate increases linearly with the square of Cu loading (if < 1.7% w/w), see Fig. 1: in fact, the authors ascribe this to the participation of Cu⁺ dimers in O₂ activation, but this argument applies only if O₂ activation is the rds, which has not been proven: see point 1) above. On the other hand, if Cu reduction is also kinetically relevant, the quadratic dependence of the SSCR rate on Cu loading suggests the participation of Cu dimers also in both the half cycles of the redox mechanism.

3) The authors ignore the mechanism of NO₂ disproportionation and heterolytic chemisorption on zeolite catalysts published years ago by Weitz and Sachtler, J Catal 131 (2005) 181. On the same topic, see also Rivallan et al. J Catal 264 (2009) 104 and Grossale et al., J Catal 256 (2008) 312.

4) Many FSCR catalytic tests were run at temperatures below the ammonium nitrate decomposition temperature (about 170 °C), which is known to result in the accumulation of ammonium nitrate on the catalyst. Did the authors check that the data were collected at steady state? Were the experiments run with increasing or decreasing temperature? Did they run a TPD to check the deposition of ammonium nitrate?

5) What is the evidence for ruling out the formation of NH₃-ligated, mobile Cu²⁺ species?

Reviewer #2 (Remarks to the Author):

In this work, the authors use a combination of reaction studies, DFT, and XAS measurements to describe the chemistry of NO₂-selective catalytic reduction over Cu-zeolite catalysts. While this work contains some interesting results about the effects of NO₂ on side-reactions that inhibit SCR reactions, I have multiple major technical concerns about the manuscript. Furthermore, some of the results are derivative of those in the literature and thus the quality and novelty of this work are insufficient for Nature Communications. Thus, I recommend this paper be rejected from Nature Communications. If the below concerns are addressed, this work could be suitable for a publication in a journal focused on catalysis and emissions control.

Technical Concerns:

- Abstract:

- The authors claim to show reactivity behavior that is “distinct from other catalyst systems”, but they do not discuss what these other systems are, nor show that the behavior in this work is different from those other systems.

- Introduction/ Discussions of literature

- The authors claim that NH₃-SCR is the most widely adopted technology to eliminate NO₂, but they should clarify that NH₃-SCR is mainly used in diesel engines.

-The authors claim that “fast SCR” occurs via NH₄NO₃ as the key intermediate. While the literature has observed that NH₄NO₃ can form during reactions involving NO₂ and NH₃ on Cu-zeolites, whether NH₄NO₃ is a reaction intermediate remains hotly debated. The authors frequently invoke NH₄NO₃ to explain their data but provide no evidence for the presence of NH₄NO₃ or its role as a reaction intermediate, thus weakening the mechanistic proposals.

- The authors claim that NO oxidation is thought to be the rate-limiting step in standard SCR. This is incorrect. Recent studies (e.g., 10.1126/science.aan5630) have shown, using kinetics, XAS, and DFT, that standard SCR occurs via the reaction of O₂ with two Cu(I) species to form binuclear intermediates (as the authors discuss later in the paper), not through NO oxidation to NO₂.

-The authors do not cite a key reference from 2012 (10.1016/j.cattod.2011.11.037) that used XAS to show that Cu(II) is the prevalent oxidation state during fast SCR.

-The authors do not adequately discuss a key reference (10.1126/science.aan5630) that studied the transient oxidation of Cu(I)(NH₃)₂ by NO₂, which is shown to be a single-site process where the first-order rate constant is independent of Cu site density, and where all Cu(I) species are oxidizable by NO₂.

- Reaction studies:

- Fig 1: the authors claim that TOFs decrease at higher Cu densities due to mass transfer limitations. This claim is vague – what species is being claimed to be mass transfer limited? Jones et al showed that under typical SCR conditions on Cu-zeolite catalysts, bed-scale mass transfer limitations do not limit standard SCR rates, and intracrystalline mass transfer limitations are unlikely to significantly affect standard SCR rates (10.1016/j.jcat.2020.05.022) . The authors must carry out the relevant experimental tests or calculate the theoretical criteria to assess the possibility of mass transfer limitations. The authors should also consider the possibility that a fraction of inactive Cu-oxides is formed at higher Cu densities.

-Concerning the increase in SCR apparent activation energy values with Cu density (Fig 1), the authors claim that this change is due to a change from Cu(I) to Cu(II) as the prevalent oxidation state (i.e., a change in kinetic regime). However, recent work (10.1016/j.jcat.2021.08.042) has shown that this change in apparent activation energies occurs even in a fixed kinetic regime.

- One of the core claims of the manuscript is that “strong oxidation by NO₂ forces Cu ions to exist mainly as fixed framework Cu²⁺ species (fw-Cu²⁺), which impede the formation of dynamic binuclear Cu+ species that serve as the main active sites for the standard SCR (SSCR) reaction”.

However, this claim is confusing because the literature has already recognized that fast SCR (with NO+NO₂) does not occur through the formation of binuclear intermediates (10.1126/science.aan5630). Since fast SCR consumes both NO and NO₂, it is a desired reaction which consumes NO, and the avoidance of binuclear intermediates alone does not explain the inhibited NO conversion. The key question, which the authors have not adequately answered, is how and why NO₂-SCR (which does not consume NO) occurs rather than fast SCR when NO₂ is present. The authors’ XAS data suggests that under both fast SCR and NO₂-SCR conditions, the most abundant surface species is framework-bound Cu(II), and thus the XAS data do not seem to distinguish the fast SCR and NO₂-SCR mechanisms.

- It was also difficult to follow the authors’ results concerning inhibition of NO-SCR, and NO₂-SCR versus fast SCR. There are two key metrics that the authors should report to clarify their results:

- (i) Comparison of NO consumption rates under fast SCR vs standard SCR conditions
- (ii) Selectivity towards standard, fast, and NO₂-SCR pathways when both NO and NO₂ are present

- Another conclusion the authors make is that NO₂-SCR occurs on Bronsted acid sites rather than Cu sites. The authors change Bronsted acid site density by changing the Cu density, which is an indirect method involving multiple changing variables. However, the authors did not carry out the critical control experiment of measuring the reaction rates on the H-form zeolite. Furthermore, on Cu-zeolite samples, the authors could independently vary the Bronsted acid site density and Cu density using ion-exchanged inert cations or by varying the Al density.

- Methods: please clarify what conversion ranges (of NO, NO₂, NH₃, etc) are used for reporting reaction rates. It is important to operate at near-differential conversions when measuring kinetics.

-Fig 2 – the authors claim that standard SCR is “severely suppressed” at low temperatures, and speculate that this has to do with pore blocking by NH₄NO₃. How can the authors rule out that the lower rates at lower temperatures are simply due to a conventional Arrhenius-type temperature dependence?

• XAS results:

-The authors claim (Fig S5a) that CuO_x species were not detected by XANES, but how would XANES distinguish these species from ion-exchanged Cu? The authors did not show comparisons with CuO_x

standards.

-The authors claim that there is “large amounts of dynamic Cu(I)(NH₃)₂” during SSCR. However, according to Fig S5. However, the pre-edge feature (Feature “B”) has a very low intensity in the XANES spectrum. In contrast, multiple literature reports indicate that a purely Cu(I)(NH₃)₂ species should have a pre-edge feature around 1.0-1.2 in the normalized XANES (e.g., 10.1002/anie.201407030, 10.1126/science.aan5630, 10.1021/cs501673g). Fig S5 seems to suggest only a small amount of Cu(I) under standard SCR conditions. In fact, the XANES for the “NO + NH₃” reduction only shows a pre-edge feature around 0.5, suggesting either that the sample re-oxidized after treatment in NO + NH₃, or that there is a large fraction of inactive Cu oxides present .

-Based on the EXAFS data, the authors argue that Cu(II) species are bound to the zeolite framework during fast SCR and NO₂-SCR conditions. Numerous literature reports (e.g., 10.1021/jacs.6b02651, 10.1021/cs501673g) have shown that both Cu(I) and Cu(II) species are NH₃-solvated under low-temperature (e.g. 473 K) NO-SCR reaction conditions, by comparison of XANES and EXAFS spectra to the spectra for homogeneous Cu-amine complexes. Computed phase diagrams have shown that Cu sites are thermodynamically favored to be NH₃-solvated under these conditions. Given all of these findings, how can the authors rationalize the claim that Cu(II) species are not NH₃-solvated under reaction conditions?

-To claim that an XAS measurement is “operando”, the authors should verify that the reactant conversion is nearly differential . This is important because the redox environment changes down the length of the catalyst bed with changing conversion. Thus, the location of the X-ray beam in the catalyst bed can greatly influence the measured XAS results if the bed is not differential (10.1021/acs.jpcclett.0c00903). Please comment on whether such effects were considered.

- DFT results:

-The authors compute various activation barriers, such as 1.27 eV for fast SCR over Bronsted acid sites. This number is one order of magnitude different than their experimentally measured apparent activation barrier under NO₂-SCR conditions (0.13 – 0.22 eV), yet the authors claim that their dataset shows that Cu sites are not involved in reactions of NO₂.

-On pg 8, the authors write “Cu(II)OH (NH₃)₂”, but the typically described species here is Cu(II)OH (NH₃)₃. Furthermore, the authors did not model Z₂Cu(II), which could be the predominant state of Cu(II) depending on the sample composition (10.1021/jacs.6b02651).

- Typos/ vague language:

-SI, page 5: the authors write “ZHO” but I believe this should be “ZNO”

-Fig S5: please clarify the temperature for each treatment.

- Page 7, the authors state "Under FSCR conditions, however, the XANES profile was more like that of pretreated Cu-SSZ-13, demonstrating that most of the copper species were Cu⁺ ions, although there was still a slight amount of Cu⁺ ions". I believe the second clause of the sentence should say "Cu²⁺ ions".

Reviewer #3 (Remarks to the Author):

This is a very well written and presented discussion of a series of measurements and calculations that provide strong support for their conclusions regarding the mechanism of inhibition of SCR activity in Cu-SSZ13 catalysts in the presence of NO₂. The work will be of interest to the growing number of researchers interested in the mechanisms of SCR activity and poisoning of copper zeolite catalysts, and the broader field of environmental catalysis. The combination of careful kinetic analysis, in situ/operando XAS studies, and DFT mechanism modeling makes a very compelling case for this conclusion. The results are applicable for the design of regeneration protocols and operational aspects of the active catalysts.

The methods used are appropriate and well described. The arguments are presented clearly and methodically. The paper should be published in Nature Communications, following a few very minor wording fixes and addressing the questions raised below.

Line 108 "generally thought to be resulted from.." should be "generally thought to result from..."

Line 251 "we found that NO₂ leads to a deep..." (need to add 'to' in this sentence)

Line 297 there is a ppm missing after NO in the mixture identification here

The experimental preparation of the catalyst resulted in zeolite with an Al/Si ratio of about 5. The XAS and kinetics measurements were carried out on samples with this ratio. In the DFT studies, the model replaced one Si atom in each double six-membered ring with Al, for a Si/Al ratio of 11. The authors should comment on this difference. Is there any reason to expect this difference to affect the conclusions about the effect of NO₂ on the SCR activity?

Otherwise, the paper is interesting, well written, and scientifically sound. It should be of interest to the readers of Nature Communications.

Point-by-Point Response to the Reviewers' Comments:

Reviewer #1:

Several questions need to be addressed in revision: 1) What is the authors' evidence for NO oxidation being the rate-determining step in the SSCR reaction at low T over Cu-SSZ-13? This is well known to be the case on V-based catalysts and on Fe-zeolites, but XAS studies show that the redox state of Cu-SSZ-13 is significantly high at SSCR conditions, which makes the Cu²⁺ reduction a kinetically relevant step.

Response: Thank you for your valuable comment. The authors agree with the reviewer that NO oxidation is the rate-determining step in the SSCR reaction on V-based catalysts and Fe-zeolites, while the situation in Cu-SSZ-13 system may be different since some researchers have found that NO oxidation occurred with difficulty over Cu-SSZ-13 (<10% below 400°C, *Catal. Lett.* 2012, 142, 295). For the SSCR reaction over Cu-SSZ-13, NO and NH₃ react together to reduce the NH₃-solvated Cu^{II} to Cu^I(NH₃)₂ species in the SCR reduction half-cycle. Then, the Cu^I(NH₃)₂ species paired in one cage to form binuclear O₂-bridged NH₃-solvated Cu^{II} complexes in the SCR oxidation half-cycle. The kinetic relevance is closely related to the O₂ pressure, reaction temperature and elemental composition of Cu-SSZ-13. In this work, the Cu-SSZ-13 samples have a high Cu loading of 3.8 wt.%, which causes Cu^{II} to be the most abundant species according to the literature (*J. Catal.*, 2020, 389, 140). Therefore, we also agree with the reviewer that Cu^{II} reduction is a kinetically relevant step in our Cu_{3.8}-SSZ-13 sample under SSCR conditions. The relevant modification is as follows:

Modification (Lines 48-51 in MS): It is generally believed that the deNO_x efficiency of the FSCR reaction should be higher than that of SSCR (reaction 4) due to bypassing the oxidation of NO, which is usually the rate-limiting step in the SSCR reaction on V-based and Fe-zeolite catalysts.^{19,20}

2) Can the authors rule out the participation of Cu²⁺ dimers in the Cu reduction mechanism of SSCR, as proposed in recent SCR literature (*ACS Catal.* 10 (2020) 5646; *Angew. Chemie* 60 (2021) 7197)? Notably, the SSCR rate increases linearly with the square of Cu loading (if < 1.7% w/w), see Fig. 1: in fact, the authors ascribe this to the participation of Cu⁺ dimers in O₂ activation, but this argument applies only if O₂ activation is the rds, which has not been proven: see point 1) above. On the other hand, if Cu reduction is also kinetically relevant, the quadratic dependence of the SSCR rate on Cu loading suggests the participation of Cu dimers also in both the half cycles of the redox mechanism.

Response: Thank you for your valuable comment. Indeed, we cannot rule out the participation of Cu^{II} dimers in the reduction half-cycle (RHC, Cu^{II}→Cu^I) based on the available data in this

manuscript. As you mentioned, the literature (*ACS Catal.*, **2020**, 10, 5646) used DFT calculations to suggest a complete reaction mechanism for low-temperature standard NH₃-SCR and found that Cu pairs existed in the RHC of Cu^{II} to Cu^I. Furthermore, a recent publication (*Angew Chem. Int. Ed.*, **2021**, 60, 7197) proposed a Cu^{II}-pair mediated low-temperature RHC pathway of SCR reaction based on experimental, kinetic fitting and computational results. Thus, the Cu^{II}-pair probably participated in the Cu reduction process. Moreover, for the oxidation half-cycle (OHC, Cu^I→Cu^{II}), the participation of Cu^I dimers has been reported by many researchers using both computational and experimental methods (*Science*, **2017**, 357, 898, *J. Am. Chem. Soc.*, **2020**, 142, 15884). In summary, Cu^{II/I} pairs probably existed in both the RHC and OHC of the SSCR reaction. Therefore, whether the Cu^I oxidation or Cu^{II} reduction is kinetically relevant, the formation of a Cu pair in the same CHA zeolite cage is indispensable for the SSCR reaction. The two publications mentioned by the reviewer are cited in the revised manuscript as Refs. 6 and 29. The relevant modification is as follows:

Modification (Lines 93-101 in MS): The SSCR rate increases linearly with the square of Cu loading when the Cu loading is below 1.7 wt.% (magnified in **Fig. 1b**), indicating the participation of Cu pairs in the standard NH₃-SCR reaction. Previous studies have reported that Cu^I dimers formed with O₂ activation in the oxidation half-cycle (Cu^I→Cu^{II}).^{4,5} Recently, Hu et al. also proposed a Cu^{II}-pair mediated low-temperature reduction half-cycle (Cu^{II}→Cu^I).⁶ Chen et al. also indicated the participation of Cu pairs in the reduction half-cycle.²⁹ Therefore, the formation of Cu pair in the same cage is significantly important for the overall standard NH₃-SCR reaction process.

3) The authors ignore the mechanism of NO₂ disproportionation and heterolytic chemisorption on zeolite catalysts published years ago by Weitz and Sachtler, *J Catal* 131 (2005) 181. On the same topic, see also Rivallan et al. *J Catal* 264 (2009) 104 and Grossale et al., *J Catal* 256 (2008) 312.

Response: Thank you for your kind reminder. We have cited these publications (Refs. 33-35) and taken the NO₂ disproportionation and heterolytic chemisorption on zeolite into consideration in the revised manuscript. The result of the present work is consistent with the reaction between NO and NH₄NO₃ proposed in the three papers. We conducted NO₂ and NH₃ co-adsorption over Cu_{2.6}-SSZ-13 sample and observed weak emission of NO, indicating the disproportionation of NO₂ (**Fig. S10**). Actually, we believe that the NO₂ disproportionation occurred at Brønsted acid sites to form NO⁺ (i.e., ZNO) and HNO₃. The ZNO would react with NH₃ to form N₂, H₂O and recover ZH. HNO₃ reacted with NH₃ to lead the formation of the intermediate NH₄NO₃, which then reacted with NO to form N₂, NO₂ and H₂O. The process is shown below:

Fig. S10 (a) NO_2 and NH_3 co-adsorption over $\text{Cu}_{2.6}\text{-SSZ-13}$ as a function of time at 120°C . (b) NO -TPSR reaction over NO_2 and NH_3 co-treated $\text{Cu}_{2.6}\text{-SSZ-13}$.

Modification

(Lines 149-151 in MS): NO_2 disproportionation occurs on the BASs to form nitrates and adsorbed NO^+ , which then react with NH_3 to form NH_4NO_3 and NH_2NO , respectively.³³⁻³⁵

(Lines 152-154 in MS): Instead, NO reacts with NH_4NO_3 at BASs to form N_2 and NO_2 through reaction (3) (TPSR shown in **Fig. S10**).

(Lines 86-93 in SI): The appearance of weak NO desorption peak indicated the disproportionation of NO_2 during NO_2 and NH_3 co-adsorption process. Actually, NO_2 disproportionation occurs on the BASs to form nitrates and adsorbed NO^+ , which then react with NH_3 to form NH_4NO_3 and NH_2NO , respectively. The consumption of NO , accompanied with the appearance of NO_2 , indicated the reaction between NO and NH_4NO_3 (reaction 3). The NO^+ could also react with NH_3 to form NH_2NO (reaction 4). The NH_2NO then easily decompose to N_2 and H_2O (reaction 5). This process was also observed in previous studies.^{6, 8-11}

4) Many FSCR catalytic tests were run at temperatures below the ammonium nitrate decomposition temperature (about 170 °C), which is known to result in the accumulation of ammonium nitrate on the catalyst. Did the authors check that the data were collected at steady state? Were the experiments run with increasing or decreasing temperature? Did they run a TPD to check the deposition of ammonium nitrate?

Response: Thank you for your valuable comment. The results of the fast SCR reaction over Cu_{2.6}-SSZ-13 as function of time at 200 °C is shown in **Fig. S5**. It can be seen that the NO, NO₂ and NH₃ concentrations decreased rapidly and then rose gradually before reaching constant levels at about ~120 minutes. In this study, we evaluated the NO_x conversion of Cu-SSZ-13 under FSCR conditions at each temperature point (<250°C) for ~240 minutes in order to guarantee steady-state conditions. Furthermore, after the FSCR reaction, we carried out TPD experiments and observed large amounts of N₂O, which indicated the occurrence of the following reaction:

Modification: (Lines 135-137 in MS): The NH₄NO₃ formation was verified by the observation of N₂O in FSCR-TPD experiment (**Fig. S5**), since the N₂O mainly originated from NH₄NO₃ decomposition.

Fig. S5 FSCR-TPD profiles over Cu_{2.6}-SSZ-13 sample. NH₃, NO, NO₂ and N₂O concentrations (a) during FSCR reaction process and (b) during TPD process of the sample treated by FSCR atmosphere.

5) What is the evidence for ruling out the formation of NH₃-ligated, mobile Cu²⁺ species?

Response: Thank you for your valuable comment. We don't rule out the formation of NH₃-ligated, mobile Cu^{II} species in this study although we proved the existence of significant framework-bonded Cu^{II} ions (fw-Cu^{II}) under FSCR conditions. Under FSCR conditions, indeed, the NH₃-solvated Cu^{II}

species exist, according to the observation of L-NH₃ (NH₃ adsorbed on Cu site) emission in the FSCR-TPD experiment (**Fig. S5**) in this work and the computed phase diagrams in the literature (*J. Am. Chem. Soc.*, 2016, 138, 6028). Nevertheless, the presence of NH₃-solvated Cu^{II} species did not influence the conclusions of this work. We calculated the diffusion of various NH₃-solvated Cu^{II} species into an adjacent cage to form Cu^{II} pairs and found that the barriers were extremely high due to the steric effect (**Fig. S15**). As a result, the formation of dynamic binuclear sites, which is required for SSCR reaction, is strongly inhibited in the presence of NO₂. We have added this discussion in the revised manuscript. Please see below.

Fig. S15 Gibbs free-energy profiles for the diffusion of various NH₃-solvated Cu^{II} species through an 8-MR window into an adjacent cage to form Cu^{II} pairs. (a) Cu^{II}(OH)(NH₃)₃; (b) Cu^{II}(NO₂)(NH₃)₃; (c) Cu^{II}(NH₃)₄. The structures of the reactants (RC), transition states (TS), and products (PC) are presented as insets. The zeolite structure is displayed by thin lines for clarity. Orange, red, blue, and white circles denote Cu, O, N, and H atoms, respectively.

Modification (Lines 244-251 in MS): Notably, although we proved the existence of significant

framework-bound Cu^{II} species under FSCR conditions, the NH₃-solvated Cu^{II} species cannot be ruled out by the XAFS experiment. Indeed, the NH₃-solvated Cu^{II} species existed, as indicated by the observation of NH₃ desorption from Cu site in FSCR-TPD profiles (**Fig. S5**), which was consistent with the computed phase diagram reported by Paolucci et al.²⁸ Nevertheless, the diffusion of various NH₃-solvated Cu^{II} species into an adjacent cage to form Cu^{II} pairs are strongly inhibited due to the extremely high barriers caused by the steric effect (**Fig S15**).

(Lines 213-217 in SI): Given the existence of NH₃-solvated species, we calculated the diffusion of various NH₃-solvated Cu^{II} species into an adjacent cage to form Cu^{II} pairs and found that the barriers were extremely high due to the steric effect. As a result, the formation of dynamic binuclear sites, which is required for high-active SSCR reaction, is strongly inhibited in the presence of NO₂.

Reviewer #2 (Remarks to the Author):

In this work, the authors use a combination of reaction studies, DFT, and XAS measurements to describe the chemistry of NO₂-selective catalytic reduction over Cu-zeolite catalysts. While this work contains some interesting results about the effects of NO₂ on side-reactions that inhibit SCR reactions, I have multiple major technical concerns about the manuscript. Furthermore, some of the results are derivative of those in the literature and thus the quality and novelty of this work are insufficient for Nature Communications. Thus, I recommend this paper be rejected from Nature Communications. If the below concerns are addressed, this work could be suitable for a publication in a journal focused on catalysis and emissions control.

Response: Thank you so much for your valuable comments. We have carefully addressed your concerns and significantly improved the manuscript and supplementary materials. As you mentioned, this work showed interesting results regarding the effects of NO₂ on side-reactions that inhibit SCR reactions. Previous studies mostly concentrated on the fast SCR reaction and ignored the effects of NO₂ on the state and coordination of active Cu sites. However, the state and coordination of Cu species in Cu-zeolites fundamentally affect the SCR reaction pathways. In this study, we demonstrated that NO₂ induces Cu species as framework fixed Cu (fw-Cu^{II}) as well as high coordination Cu(NH₃)₄ (or Cu(NH₃)₃NO₂), the mobility of which was inhibited due to electrostatic force and steric effects. The highly active Cu pairs that play key roles in SSCR reaction were therefore difficult to form. As a result, the conversion of NO was unexpectedly inhibited in the presence of NO₂. This was strikingly distinct from the NO₂-accelerated NO_x conversion over Fe-based zeolite or V-based oxide catalysts. We believe that this finding will arouse widespread interest and attention from the readership of *Nature Communications*.

➤ **Technical Concerns:**

• **Abstract:**

- 1) The authors claim to show reactivity behavior that is “distinct from other catalyst systems”, but they do not discuss what these other systems are, nor show that the behavior in this work is different from those other systems.

Response: Thank you for your valuable comment. Generally, the deNO_x efficiency of Fe-based zeolites and most oxide SCR catalysts increases with the addition of NO₂ to the SCR feed. However, an obvious inhibition of NO_x conversion by NO₂ was observed over the Cu-SSZ-13 catalyst. For comparison, we carried out the SSCR and FSCR reactions over the Cu_{2.6}-SSZ-13, Fe_{2.0}/Beta, and V- and Ce-based oxide SCR catalysts (**Fig. S3**). The results clearly showed that NO₂ addition inhibited

the NO_x conversion over Cu-SSZ-13 while significantly promoting the NO_x conversion over other catalyst systems. We have added the SSCR and FSCR activity of Cu-SSZ-13, Fe/Beta, and V- and Ce-based oxide catalysts in the revised manuscript. The relevant modification is as follows:

Fig. S3 NO_x conversion of Cu-SSZ-13, Fe/Beta, VWTiO_x and CeWSnO_x catalysts under FSCR and SSCR conditions. FSCR conditions: [NO]=[NO₂]=250 ppm, [NH₃]=500 ppm, [O₂]=5 vol.% [H₂O]=3.5 vol.%. SSCR conditions: [NO]=500 ppm, [NH₃]=500 ppm, [O₂]=5 vol.% [H₂O]=3.5 vol.%. The GHSVs were 800,000, 400,000, 200,000 and 400,000 h⁻¹, respectively. The Cu and Fe contents of Cu-SSZ-13 and Fe/Beta were 2.6 and 2.0 wt.%, respectively. The molar ratio of Ce:W:Sn in the CeWSnO_x catalyst was 1:0.2:2. The V and W contents for VWTiO_x were 0.5 and 7.5 wt.%, respectively, and the support was anatase TiO₂.

Modification (Lines 16-20 in MS): Commercial Cu-exchanged small-pore SSZ-13 (Cu-SSZ-13) zeolite catalysts are highly active for the standard selective catalytic reduction (SCR) of NO with NH₃. However, their activity is unexpectedly inhibited in the presence of NO₂ at low temperatures. This was strikingly distinctive from the NO₂-accelerated NO_x conversion over other typical SCR catalyst systems.

(Lines 124-128 in MS): Then, the FSCR reaction over Cu-SSZ-13 was carried out as shown in **Fig. S2a** and **Fig. S3a**. Compared to the SSCR reaction, the NO_x conversion over Cu-SSZ-13 was significantly inhibited in the presence of NO₂, which was strikingly distinct from the NO₂-accelerated NO_x conversion over Fe-based zeolite and oxide catalysts (**Fig. S3**).

• **Introduction/ Discussions of literature**

- 2) The authors claim that NH₃-SCR is the most widely adopted technology to eliminate NO_x, but they should clarify that NH₃-SCR is mainly used in diesel engines.

Response: Thank you for your suggestion. The relevant modification is as follows:

Modification (Lines 35-37 in MS): Selective catalytic reduction of NO_x with NH₃ (NH₃-SCR) is the most widely adopted technology for the removal of NO_x from diesel engines.^{1,2}

- 3) The authors claim that “fast SCR” occurs via NH₄NO₃ as the key intermediate. While the literature has observed that NH₄NO₃ can form during reactions involving NO₂ and NH₃ on Cu-zeolites, whether NH₄NO₃ is a reaction intermediate remains hotly debated. The authors frequently invoke NH₄NO₃ to explain their data but provide no evidence for the presence of NH₄NO₃ or its role as a reaction intermediate, thus weakening the mechanistic proposals.

Response: Thank you for your valuable comments. We carried out FSCR-TPD and NO-TPSR after NO₂ and NH₃ treatment to prove the formation of NH₄NO₃ under FSCR conditions. As shown in **Fig. S5b**, the large amounts of N₂O emission caused by the decomposition of NH₄NO₃ indicated the formation of NH₄NO₃ during the reaction process under FSCR conditions.

In addition, we carried out NO₂ and NH₃ co-adsorption over the Cu-SSZ-13 sample, after which the NO-TPSR was conducted (**Fig. S10**). The formation of NO₂ indicated the reaction between NO and NH₄NO₃.

Therefore, the NH₄NO₃ is an important intermediate under FSCR conditions and can react with NO to complete the FSCR reaction cycle. Moreover, NH₄NO₃ formation was also observed in many publications (*J. Phys. Chem. C*, **2021**, 125, 21975; *ACS Catal.*, **2020**, 10, 2334; *J. Catal.*, **2015**, 329, 490.)

Fig. S5 FSCR-TPD profiles over Cu_{2.6}-SSZ-13 sample. NH₃, NO, NO₂ and N₂O concentrations (a) during FSCR reaction process and (b) during TPD process of the sample treated by FSCR atmosphere.

Fig. S10 (a) NO₂ and NH₃ co-adsorption over Cu_{2.6}-SSZ-13 as a function of time. (b) NO-TPSR reaction over NO₂ and NH₃ co-treated Cu_{2.6}-SSZ-13.

Modification:

(Lines 135-137 in MS): The NH₄NO₃ formation was verified by the observation of N₂O in an FSCR-TPD experiment (**Fig. S5**), since the N₂O mainly originated from NH₄NO₃ decomposition.

(Lines 149-151 in MS): NO₂ disproportionation occurs on the BASs to form nitrates and adsorbed NO⁺, which then react with NH₃ to form NH₄NO₃ and NH₂NO, respectively.³³⁻³⁵

(Lines 152-154 in MS): Instead, NO reacts with NH₄NO₃ at BASs to form N₂ through reaction (3) (TPSR shown in **Fig. S10**).

(Lines 86-93 in SI): The appearance of a weak NO desorption peak indicated the disproportionation

of NO₂ during the NO₂ and NH₃ co-adsorption process. Actually, NO₂ disproportionation occurs on the BASs to form nitrates and adsorbed NO⁺, which then react with NH₃ to form NH₄NO₃ and NH₂NO, respectively. The consumption of NO, accompanied by the appearance of NO₂, indicated the reaction between NO and NH₄NO₃ (reaction 3). NO⁺ could also react with NH₃ to form NH₂NO (reaction 4). NH₂NO then easily decomposes to N₂ and H₂O (reaction 5). This process was also observed in previous studies.^{6, 8-11}

- 4) The authors claim that NO oxidation is thought to be the rate-limiting step in standard SCR. This is incorrect. Recent studies (e.g., 10.1126/science.aan5630) have shown, using kinetics, XAS, and DFT, that standard SCR occurs via the reaction of O₂ with two Cu(I) species to form binuclear intermediates (as the authors discuss later in the paper), not through NO oxidation to NO₂.

Response: Thank you for your valuable comment. We agree with the reviewer that NO oxidation is not the rate-limiting step in standard SCR over Cu-SSZ-13. In fact, NO oxidation is generally the rate-determining step in the SSCR reaction on V-based catalysts and Fe-zeolites, and the situation in Cu-SSZ-13 system may be different since some researchers have found that NO oxidation occurred with difficulty over Cu-SSZ-13 (<10% below 400°C, *Catal. Lett.* 2012, 142, 295). The relevant modification is as follows:

Modification (Lines 48-51 in MS): It is generally believed that the deNO_x efficiency of the FSCR reaction should be higher than that of SSCR (reaction 4) due to bypassing NO oxidation, which is usually the rate-limiting step in the SSCR reaction on V-based and Fe-zeolite catalysts.^{19,20}

- 5) The authors do not cite a key reference from 2012 (10.1016/j.cattod.2011.11.037) that used XAS to show that Cu(II) is the prevalent oxidation state during fast SCR.

Response: Thank you for your kind reminder. We have cited this reference in the revised manuscript (Ref. 26) and supporting information file. The result in this reference confirmed that Cu^{II} is the prevalent oxidation state under FSCR conditions compared to that under SSCR conditions.

Modification (Lines 63-66 in MS): McEwen et al. found that four-fold-coordinated Cu(II) species

dominate the Cu-SSZ-13 catalyst under FSCR conditions, which differs from the composition under SSCR conditions, where Cu(I) and Cu(II) species both exist.²⁶

(Lines 235-237 in MS): In the presence of NO₂, however, the CNs of the first shell significantly increased, indicating the oxidation of copper species, which was also supported by the results of McEwen et al.²⁶

(Lines 165-167 in SI): This result suggests that Cu²⁺ is the prevalent oxidation state under FSCR conditions, which is consistent with the results reported by McEwen et al.²³

6) The authors do not adequately discuss a key reference (10.1126/science.aan5630) that studied the transient oxidation of Cu^I(NH₃)₂ by NO₂, which is shown to be a single-site process where the first-order rate constant is independent of Cu site density, and where all Cu^I species are oxidizable by NO₂.

Response: Thank you for your kind reminder. This reference showed important progress in the SSCR mechanism over Cu-SSZ-13, which included the formation of dynamic and reversible multinuclear sites. Also, the authors of the reference discussed the oxidation process of Cu^I(NH₃)₂ species by O₂ and NO₂. Different from the second-order oxidation by O₂, the transient oxidation of Cu^I(NH₃)₂ by NO₂ was shown to be a first-order dependent, which indicated that NO₂ probably changed the SCR reaction active sites from dimer Cu to isolated Cu species. Moreover, NO₂ can oxidize the residual Cu^I(NH₃)₂ species that cannot be oxidized by O₂, which is consistent with our results as shown in **Fig. 3**. Although the reference mentioned that NO₂ accelerates SCR rates by accelerating Cu^I oxidation kinetics, the sacrifice of the mobility of Cu species due to its high valence and coordination number was not considered, which would significantly affect the SSCR reaction. For the first time, our study reported that NO₂ inhibited the mobility of Cu by oxidizing the Cu species into a high-valence state.

In addition, we calculated the FSCR reaction pathways over various fw-Cu^{II} and NH₃-solvated Cu^{II} species that may exist under FSCR conditions and found that the energy barriers of all the reaction pathways are higher than that of the SSCR reaction over dynamic binuclear Cu sites. Based on the above considerations, we further discuss this key reference in the revised manuscript. The relevant modification is as follows:

Modification (Lines 66-68 in MS): Paolucci et al. investigated the oxidation process of Cu^I(NH₃)₂ species by O₂ and NO₂. It was found that oxidation by NO₂ occurred at isolated Cu sites, rather than at the Cu dimer sites required for O₂ activation.⁵

(Lines 219-229 in MS): This phenomenon is consistent with the result reported by Paolucci et al. showing that NO_2 can oxidize the residual $\text{Cu}^{\text{I}}(\text{NH}_3)_2$ species that cannot be oxidized by O_2 . As also reported by Paolucci et al., the transient oxidation of $\text{Cu}^{\text{I}}(\text{NH}_3)_2$ species by NO_2 is a single-site process without formation of Cu dimers. Therefore, it can be inferred that the presence of NO_2 probably changed the SCR reaction active sites from dimer Cu to isolated Cu species, which further influence the SSCR reaction. This deduction indicated that most Cu species are bonded with the zeolite framework and that the mobility of Cu species is limited during the process of $\text{Cu}^{\text{I}}(\text{NH}_3)_2$ oxidation by NO_2 . Although the transient reaction can reflect the Cu state and coordination during half-cycles, it was deemed more meaningful to identify the Cu species under FSCR reaction conditions.

(Lines 143-150 in SI): This phenomenon is consistent with the result reported by Paolucci et al. that the $\text{Cu}^{\text{I}}(\text{NH}_3)_2$ oxidation by O_2 has a theoretical limit, while NO_2 can fully oxidize the $\text{Cu}^{\text{I}}(\text{NH}_3)_2$ species. Moreover, it was found that the transient oxidation of $\text{Cu}^{\text{I}}(\text{NH}_3)_2$ species by NO_2 is a single-site process where the first-order rate constant is independent of Cu site density. However, it was known that dimer Cu species participated in SSCR reaction. The existence of NO_2 probably changed the active sites of SCR reaction from dimer Cu to isolated Cu species, which further influence the SSCR reaction.

• **Reaction studies:**

7) Fig 1: the authors claim that TOFs decrease at higher Cu densities due to mass transfer limitations. This claim is vague – what species is being claimed to be mass transfer limited? Jones et al showed that under typical SCR conditions on Cu-zeolite catalysts, bed-scale mass transfer limitations do not limit standard SCR rates, and intracrystalline mass transfer limitations are unlikely to significantly affect standard SCR rates (10.1016/j.jcat.2020.05.022). The authors must carry out the relevant experimental tests or calculate the theoretical criteria to assess the possibility of mass transfer limitations. The authors should also consider the possibility that a fraction of inactive Cu-oxides is formed at higher Cu densities.

Response: Thank you for your valuable comments. We first excluded the formation of inactive Cu-oxides at high Cu densities. As shown in **Fig S12**, we compared the XANES and EXAFS profiles of hydrated $\text{Cu}_{3.8}\text{-SSZ-13}$, Cu_2O and CuO . There are no observable features at 8982 and 8986 eV in the XANES profile of $\text{Cu}_{3.8}\text{-SSZ-13}$, which can be attributed to the signals of Cu_2O and CuO species, respectively. Moreover, the absence of Cu-Cu scattering at 2-3 Å also indicated there are no CuO_x species in the $\text{Cu}_{3.8}\text{-SSZ-13}$ sample.

Fig. S12 Cu K-edge XANES and EXAFS profiles of CuO, Cu₂O and hydrated Cu_{3.8}-SSZ-13 samples.

As for the TOF decrease, we found that the NO conversion over Cu_{2.6}-SSZ-13 and Cu_{3.8}-SSZ-13 at 180 °C was relatively high and beyond the differential regime interval. So, we deleted the TOF change at 180 °C. Nevertheless, we still found a weak decrease in the TOF at high Cu loading. There are some possible reasons for this phenomenon. First, the article you referenced (10.1016/j.jcat.2020.05.022; Ref. 30) showed a detailed kinetic study of Cu^{II} reduction and dual-site Cu^I oxidation process. However, we noted that Cu-SSZ-13 with Si/Al of 15 and maximum Cu loading of 2.47 wt.% was used, which is much different from our work (Si/Al of 5 and maximum Cu loading of 3.8 wt.%). In this work, every 2.4 and 3.5 CHA cages contain one Cu ion for Cu_{3.8}-SSZ-13 and Cu_{2.6}-SSZ-13 samples, respectively (according to the computing method in 10.1016/j.jcat.2020.05.022). As a result, the formed Cu-NH₃ complex as well as some oxygen-bridged dimeric species under SSCR conditions at low temperatures probably occupy CHA cages and block pore openings. The reactants approach the formed intermediates and are consumed, which leads to the underutilization of the Cu ions deep in the pores. The decrease of TOF in high Cu-loading Cu-SSZ-13 was also reported by Gao et al. (*J. Catal.*, 2014, 319, 1-14). Second, we believe that the Si/Al ratio (5 in this work) is also an influencing factor. The Al-rich condition is more likely to accelerate the mobility of Cu ions and form oxygen-bridged dimeric species since the Al site probably plays a “springboard” role for the Cu-NH₃ complex under SSCR conditions. Study on the effect of Al on the transfer of the Cu-NH₃ complex is ongoing in our group but beyond the scope of the present work. Besides, the Al distribution and synthetic method (the *in-situ* synthesis method was used in this work) can also influence the distribution and further utilization efficiency of Cu ions. For these reasons, we believe that it was reasonable for the TOF decrease at high Cu-loaded Cu-SSZ-13.

The goal of the present work is to uncover the distinctive SCR behavior over Cu-SSZ-13 in the presence of NO₂. We conducted SSCR reactions in order to compare the NO conversion under different conditions. To avoid ambiguity according to your comment, we modified the related sentences.

Modification (Lines 104-109 in MS): At high Cu loading, however, the decline of TOF is probably due to the underutilization of the active Cu sites. According to the calculation method in the literature reported by Jones et al.,³⁰ every 2.4 and 3.5 CHA cages contain one Cu ion for Cu_{3.8}-SSZ-13 and Cu_{2.6}-SSZ-13 samples, respectively. The formed Cu-NH₃ complex or dimer Cu species under SSCR conditions probably impede the access of reactants to the Cu ions deep inside the pores, causing inefficiency in the use of Cu ions.³¹

8) Concerning the increase in SCR apparent activation energy values with Cu density (Fig 1), the authors claim that this change is due to a change from Cu(I) to Cu(II) as the prevalent oxidation state (i.e., a change in kinetic regime). However, recent work (10.1016/j.jcat.2021.08.042) has shown that this change in apparent activation energies occurs even in a fixed kinetic regime.

Response: Thank you for your valuable comments. We have noted that the increase of Ea with increasing Cu density does not solely reflect a change in kinetic regime according to the work you referenced (10.1016/j.jcat.2021.08.042; Ref. 32). The Ea of Cu^I oxidation can also increase monotonically with Cu density in a fixed kinetic regime due to the non-mean-field behavior of Cu-SSZ-13 in the NH₃-SCR reaction. We also noted that increasing the Cu density would increase the kinetic relevance of Cu^{II} reduction, the Ea of which was higher than the Ea of Cu^I oxidation. Therefore, we thought that both factors lead to the increasing Ea of the SSCR reaction in this work. The relevant modification is as follows:

Modification (Lines 112-119 in MS): Recently, Krishna et al. reported that the Ea of Cu^I oxidation increased monotonically with Cu density in a fixed kinetic regime due to the non-mean-field behavior of Cu-SSZ-13 in the NH₃-SCR reaction and that the Ea of Cu^{II} reduction was unchanged when the Cu loading was higher than 0.69 wt.%.³² On the other hand, the kinetic relevance of Cu^{II} reduction increased with increasing Cu density, the Ea of which was higher than that of Cu^I oxidation.^{30,32} Therefore, the increase of the Ea in Cu^I oxidation and the kinetic relevance of Cu^{II} reduction contributed to the increase in the Ea of the SSCR reaction.

9) One of the core claims of the manuscript is that “strong oxidation by NO₂ forces Cu ions to exist mainly as fixed framework Cu²⁺ species (fw-Cu²⁺), which impede the formation of dynamic binuclear Cu⁺ species that serve as the main active sites for the standard SCR (SSCR) reaction”.

However, this claim is confusing because the literature has already recognized that fast SCR (with NO+NO₂) does not occur through the formation of binuclear intermediates (10.1126/science.aan5630). Since fast SCR consumes both NO and NO₂, it is a desired reaction which consumes NO, and the avoidance of binuclear intermediates alone does not explain the inhibited NO conversion. The key question, which the authors have not adequately answered, is how and why NO₂-SCR (which does not consume NO) occurs rather than fast SCR when NO₂ is present. The authors' XAS data suggests that under both fast SCR and NO₂-SCR conditions, the most abundant surface species is framework-bound Cu^(II), and thus the XAS data do not seem to distinguish the fast SCR and NO₂-SCR mechanisms.

Response: Thank you for your valuable comments. Although the *Science* paper recognized that the fast SCR (with NO+NO₂) does not occur through the formation of binuclear intermediates, the reason for this phenomenon was not clearly answered, and this is what our work sought to explore.

First, regarding the SSCR reaction pathway over Cu-SSZ-13, it was reported that NO reacts with NH₃-solvated Cu^{II} species to form dynamic Cu^I(NH₃) species in the reduction half-cycle. The formed dynamic Cu^I(NH₃) species diffuse into the same cage and pair to activate O₂, which was thought to be the rate-determining step of the SSCR reaction. The energy barrier of this process was about 1.0 eV (ACS Catal. 2020, 10, 5646). However, in the presence of NO₂, the strong oxidation by NO₂ forces Cu ions to exist mainly as Cu^{II} (fw-Cu^{II} or NH₃-solvated Cu^{II} species). The various Cu^{II} species have more difficulty diffusing into the same cage and facilitating O₂ activation to complete the SSCR reaction than Cu^I(NH₃)₂ species, due to the strong interaction with the zeolite framework and steric effects (see **Fig. S15**). As a result, the SSCR reaction with the participation of dimeric Cu was significantly inhibited.

Then, the mononuclear sites are dominant over Cu-SSZ-13 in the presence of NO₂. However, the energy barrier of the SSCR reaction occurring at a single Cu site is higher than that of the SSCR reaction with dimeric Cu as an intermediate. Gao et al. calculated the oxidation of solitary Cu^I(NH₃)₂ by O₂ and NO and found a higher energy barrier of 175 kJ/mol (Gao et al., *J. Am. Chem. Soc.*, 2017, 139, 4935.).

Next, we turn to the SCR reaction in the presence of both NO and NO₂. The formation of NH₄NO₃ by the reaction between NO₂ and NH₃ was proved by FSCR-TPD (**Fig. S5**). Therefore, we calculated the FSCR reaction pathway including the formation of NH₄NO₃ at single Cu sites as well as BASs. The reaction at BASs showed a lower energy barrier of 1.27 eV than that at the Cu site. Nevertheless, the energy barrier is still higher than that of the SSCR reaction with the participation of

dimeric Cu. Therefore, the NO conversion was inhibited under FSCR conditions compared to that under SSCR conditions in the Cu-SSZ-13 system. NO₂, however, undergoes facile reaction with NH₃ to form NH₄NO₃ at BASs. As a result, the conversion of NO₂ is higher than that of NO.

Lastly, XAFS cannot distinguish the FSCR and NO₂-SCR mechanisms well. However, this did not influence our conclusion since we just wanted to prove that the state and coordination of Cu species in the presence of NO₂ are different from those under SSCR conditions (only NO as NO_x) by using WT-EXAFS. As shown in **Fig. S13c**, there is only a slight difference, in that a small amount of Cu^I species (feature B) existed under FSCR conditions and the Cu^I species were completely oxidized under NO₂-SCR conditions.

In summary, NO₂ forces Cu ions to exist as solitary Cu^{II} (fw-Cu^{II} or NH₃-solvated Cu^{II} species), which limits the formation of dynamic binuclear Cu^I species to complete the SSCR reaction cycle. However, the Ea of the SSCR and FSCR reaction at solitary Cu^{II} sites are both higher than that of the SSCR reaction with the participation of Cu dimers, leading to the inhibition effect on NO conversion.

10) It was also difficult to follow the authors' results concerning inhibition of NO-SCR, and NO₂-SCR versus fast SCR. There are two key metrics that the authors should report to clarify their results:

- (i) Comparison of NO consumption rates under fast SCR vs standard SCR conditions
- (ii) Selectivity towards standard, fast, and NO₂-SCR pathways when both NO and NO₂ are present

Response: Thank you for your valuable suggestions. We compared the NO consumption rates under SSCR and FSCR conditions as shown in **Fig. S4**. It was clearly seen that the NO conversion was suppressed under FSCR conditions. In addition, we depicted the NO and NO₂ conversion under SSCR, FSCR and NO₂-SCR conditions as shown in **Fig. S11**. The NO conversion under SSCR conditions was remarkably higher than that under FSCR conditions, which indicated that the SSCR reaction rarely occurs under FSCR conditions. The NO₂ conversion under NO₂-SCR conditions was a little higher than that under FSCR conditions. We ascribed the low NO conversion to the FSCR reaction, which consumed both NO and NO₂, since the NO conversion would be relatively higher if the SSCR reaction occurred.

Fig. S4 Comparison of NO reaction rates as a function of Cu loading over Cu-SSZ-13 catalysts under SS CR and FSCR conditions at 150 and 180°C.

Fig. S11 The NO and NO₂ conversion over Cu_{2.6}-SSZ-13 catalyst under SS CR, FSCR and NO₂-SCR conditions.

Modification (Lines 131-133 in MS): The NO consumption rates under FSCR and SS CR conditions were compared (**Fig. S4**) and the result showed that NO reduction was severely suppressed at low temperatures under FSCR conditions.

(Lines 154-160 in MS): Furthermore, the NO and NO₂ conversion levels over Cu_{2.6}-SSZ-13 under SS CR, FSCR and NO₂-SCR conditions are separately depicted in **Fig. S11**. The NO conversion under SS CR conditions was remarkably higher than that under FSCR conditions, which indicated

that the SSCR reaction was significantly inhibited under FSCR conditions. We ascribed the low NO conversion to the reaction with NH_4NO_3 (i.e., FSCR reaction) and the extra NO_2 conversion to the reaction between NO_2 and NH_3 .

11) Another conclusion the authors make is that NO_2 -SCR occurs on Bronsted acid sites rather than Cu sites. The authors change Bronsted acid site density by changing the Cu density, which is an indirect method involving multiple changing variables. However, the authors did not carry out the critical control experiment of measuring the reaction rates on the H-form zeolite. Furthermore, on Cu-zeolite samples, the authors could independently vary the Bronsted acid site density and Cu density using ion-exchanged inert cations or by varying the Al density.

Response: Thank you for your valuable comments. According to your suggestion, we carried out the NO_2 -SCR reaction over H-SSZ-13 and $\text{Cu}_{2.6}$ -SSZ-13 with Si/Al of 6, 12 and 18. As expected, the H-SSZ-13 and $\text{Cu}_{2.6}$ -SSZ-13 catalysts with low Si/Al of 6 have the highest NO_2 conversion due to their high density of Brønsted acid sites at low temperatures. We have added this result to the manuscript. Please see below.

Fig. S9 NO_x conversion over H-SSZ-13 and $\text{Cu}_{2.6}$ -SSZ-13 with Si/Al of 6, 12 and 18 under NO_2 -SCR conditions.

Modification (Lines 144-147 in MS): Moreover, we carried out the NO_2 -SCR reaction over H-SSZ-13 and $\text{Cu}_{2.6}$ -SSZ-13 with different Si/Al ratios and found that the zeolites with low Si/Al exhibited high NO_x conversion due to their high number of BASs at low temperatures (**Fig. S9**).

Methods:

12) please clarify what conversion ranges (of NO, NO_2 , NH_3 , etc) are used for reporting reaction rates. It is important to operate at near-differential conversions when measuring kinetics.

Response: Thank you for your valuable comments. We added the NO_x conversion over various Cu-SSZ-13 samples under SSCR reactions condition in **Fig. S1**. For low Cu-loaded samples (**Fig. S1a**), the NO_x conversion levels below 160 °C are in a near-differential state. For high Cu-loaded samples (**Fig. S1b and S1c**), however, we have to use different weights to achieve the near-differential conversions in the temperature range between 120 to 160 °C. Under FSCR and NO₂-SCR reaction conditions, the NO, NO₂ and NO_x conversion levels are depicted in **Fig. S2** and **Fig. S7**. Except for the Cu_{0.4}-SSZ-13 sample that exhibited only slightly high NO₂ conversion (less than 25%), the NO, NO₂ and NO_x conversion levels of all other samples are less than 20%, which was near-differential, which rationalized the kinetic studies. The kinetic parameter of NH₃ was not necessary in this study.

Fig. S1 NO_x conversion over Cu-SSZ-13 with different Cu loadings under SSCR conditions. SSCR conditions: [NO]=500 ppm, [NH₃]=500 ppm, [O₂]=5 vol.% [H₂O]=3.5 vol.%, Total flow=500 mL/min.

13) Fig 2 – the authors claim that standard SCR is “severely suppressed” at low temperatures, and speculate that this has to do with pore blocking by NH₄NO₃. How can the authors rule out that the lower rates at lower temperatures are simply due to a conventional Arrhenius-type temperature dependence?

Response: Thank you for your valuable comments. We want to indicate that the SSCR reaction was severely suppressed in the presence of NO₂ (as shown in Fig. S4) rather than that SSCR reaction was suppressed at low temperatures (compared to high temperatures). We depicted the NO conversion over Cu-SSZ-13 catalysts when both NO and NO₂ are present. It can be seen that the NO conversion under SSCR conditions was significantly higher than that in the co-existence of NO and NO₂ at the same temperature. Therefore, the inhibition is due to the different reaction mechanisms rather than

the temperature.

Fig. S4 Comparison of NO reaction rates as a function of Cu loading over Cu-SSZ-13 catalysts under SSCR and FSCR conditions at 150 and 180°C.

• **XAS results:**

14) The authors claim (Fig. S5a) that CuO_x species were not detected by XANES, but how would XANES distinguish these species from ion-exchanged Cu? The authors did not show comparisons with CuO_x standards.

Response: Thank you for your valuable comments. We compared the XANES and EXAFS profiles of hydrated $\text{Cu}_{3.8}$ -SSZ-13, Cu_2O and CuO (**Fig. S12**). There are no observable features at 8982 and 8986 eV in the XANES profile of $\text{Cu}_{3.8}$ -SSZ-13, which would be attributed to the signals of Cu_2O and CuO species, respectively. Moreover, the absence of Cu-Cu scattering at 2-3 Å in EXAFS profiles also indicated that no CuO_x species are present in $\text{Cu}_{3.8}$ -SSZ-13 sample. We have added **Fig. S12** in Supporting Information.

Fig. S12 Cu K-edge XANES and EXAFS profiles of CuO, Cu₂O and hydrated Cu_{3.8}-SSZ-13 samples.

15) The authors claim that there is “large amounts of dynamic Cu^(I)(NH₃)₂” during SSCR. However, according to Fig S5. However, the pre-edge feature (Feature “B”) has a very low intensity in the XANES spectrum. In contrast, multiple literature reports indicate that a purely Cu^(I)(NH₃)₂ species should have a pre-edge feature around 1.0-1.2 in the normalized XANES (e.g., 10.1002/anie.201407030, 10.1126/science.aan5630, 10.1021/cs501673g). Fig S5 seems to suggest only a small amount of Cu^(I) under standard SCR conditions. In fact, the XANES for the “NO + NH₃” reduction only shows a pre-edge feature around 0.5, suggesting either that the sample re-oxidized after treatment in NO + NH₃, or that there is a large fraction of inactive Cu oxides present.

Response: Thank you for your valuable comments. Compared to the literature, the pre-edge feature (Feature B in **Fig. S13**) of Cu-SSZ-13 seems to be a little lower in this work. There are some possible reasons for this phenomenon. First, the high Cu loading (3.8 wt.%) of Cu-SSZ-13 in this work makes Cu species more difficult to reduce compared to the low-Cu loaded Cu-SSZ-13. In the *Science* paper (10.1126/science.aan5630), for instance, the Cu^I fraction in the Cu-SSZ-13 catalyst with Si/Al of 4.5 and Cu loading of 3.7 wt.% is only 10% in the SSCR operando steady state. Second, the *in-situ* XAFS spectra were collected in the fluorescence pattern mode when the samples were exposed to the reactants, which was different from the studies that used transmission mode in *operando* situations. We apologize that we confused the *in-situ* and *operando* conditions. In this study, the samples are only exposed *in-situ* to the SCR reaction atmosphere in a reaction chamber rather than in an *operando* tube reactor. We have revised the related text in the manuscript. In addition, we have excluded the formation of inactive Cu-oxides at high Cu densities as stated in the

response for the aforementioned comment.

Modification (Lines 242-244 in MS): The above results proved the existence of greater amounts of dynamic $\text{Cu}^{\text{I}}(\text{NH}_3)_2$ species under SSQR reaction conditions than that under FSCR and NO_2 -SCR reaction conditions.

16) -Based on the EXAFS data, the authors argue that $\text{Cu}(\text{II})$ species are bound to the zeolite framework during fast SCR and NO_2 -SCR conditions. Numerous literature reports (e.g., 10.1021/jacs.6b02651, 10.1021/cs501673g) have shown that both $\text{Cu}(\text{I})$ and $\text{Cu}(\text{II})$ species are NH_3 -solvated under low-temperature (e.g. 473 K) NO -SCR reaction conditions, by comparison of XANES and EXAFS spectra to the spectra for homogeneous Cu-amine complexes. Computed phase diagrams have shown that Cu sites are thermodynamically favored to be NH_3 -solvated under these conditions. Given all of these findings, how can the authors rationalize the claim that $\text{Cu}(\text{II})$ species are not NH_3 -solvated under reaction conditions?

Response: Thank you for your valuable comment. Basically, the publications mentioned by the reviewer reported that both Cu^{I} and Cu^{II} species are NH_3 -solvated at low temperatures. However, they did not consider the presence of NO_2 , which has strong oxidation ability that may favor the oxidation of Cu^{I} species. Nevertheless, we cannot rule out the formation of NH_3 -solvated Cu^{2+} species in this study although we proved the existence of significant framework-bonded Cu^{2+} ions (fw- Cu^{II}) under FSCR conditions by the *in-situ* EXAFS data. Under FSCR conditions, indeed, the NH_3 -solvated Cu^{II} species did exist, according to the observation of L- NH_3 (NH_3 adsorbed on Cu site) emission in the FSCR-TPD experiment (**Fig. S5**) and computed phase diagrams in the literature (*J. Am. Chem. Soc.*, 2016, 138, 6028). Therefore, both fw- Cu^{II} species and NH_3 -solvated Cu^{II} species may exist under FSCR conditions.

Nevertheless, the presence of NH_3 -solvated Cu^{II} species did not influence the conclusion of this work. We calculated the diffusion of various NH_3 -solvated Cu^{II} species into an adjacent cage to form Cu^{II} pairs and found that the barriers were extremely high due to the steric effect (**Fig. S15**). As a result, the formation of dynamic binuclear sites, which is required for the SSQR reaction, is strongly inhibited in the presence of NO_2 . We have added this discussion in the revised manuscript. Please see below.

Fig. S5 FSCR-TPD profiles over Cu_{2.6}-SSZ-13 sample. NH₃, NO, NO₂ and N₂O concentrations (a) during FSCR reaction process and (b) during TPD process of the sample treated by FSCR atmosphere.

Fig. S15 Gibbs free energy profiles for the diffusion of various NH₃-solvated Cu^{II} species through an 8-MR window into an adjacent cage to form Cu^{II} pairs. (a) Cu^{II}(OH)(NH₃)₃; (b) Cu^{II}(NO₂)(NH₃)₃; (c) Cu^{II}(NH₃)₄. The structures of the reactants (RC), transition states (TS), and products (PC) are presented as insets. The zeolite structure is displayed by thin lines for clarity. Orange, red, blue, and

white circles denote Cu, O, N, and H atoms, respectively.

Modification (Lines 244-251 in MS): Notably, although we proved the existence of significant framework-bound Cu^{II} species under FSCR conditions, the NH₃-solvated Cu^{II} species cannot be ruled out by the XAFS experiment. Indeed, the NH₃-solvated Cu^{II} species existed, as indicated by the observation of NH₃ desorption from Cu sites in FSCR-TPD profiles (**Fig. S5**), which was consistent with the computed phase diagram reported by Paolucci et al.²⁸ Nevertheless, the diffusion of various NH₃-solvated Cu^{II} species into an adjacent cage to form Cu^{II} pairs are strongly inhibited due to the extremely high barriers caused by the steric effect (**Fig S15**).

(Lines 213-217 in SI): Given the existence of NH₃-solvated species, we calculated the diffusion of various NH₃-solvated Cu^{II} species into an adjacent cage to form Cu^{II} pairs and found that the barriers were extremely high due to the steric effect. As a result, the formation of dynamic binuclear sites, which is required for high-active SSCR reaction, is strongly inhibited in the presence of NO₂.

17) -To claim that an XAS measurement is “operando”, the authors should verify that the reactant conversion is nearly differential. This is important because the redox environment changes down the length of the catalyst bed with changing conversion. Thus, the location of the X-ray beam in the catalyst bed can greatly influence the measured XAS results if the bed is not differential (10.1021/acs.jpcclett.0c00903). Please comment on whether such effects were considered.

Response: Thank you for your valuable comments. We apologize that we confused the *in-situ* and *operando* conditions. In this study, the samples are only exposed *in-situ* to the SCR reaction atmosphere in a reaction chamber rather than in an *operando* tube reactor. Therefore, we have deleted the word “*operando*” in the manuscript.

• **DFT results:**

18) The authors compute various activation barriers, such as 1.27 eV for fast SCR over Brønsted acid sites. This number is one order of magnitude different than their experimentally measured apparent activation barrier under NO₂-SCR conditions (0.13 – 0.22 eV), yet the authors claim that their dataset shows that Cu sites are not involved in reactions of NO₂.

Response: Thank you for your valuable comment. The fast SCR reaction we calculated involved the reaction of both NO and NO₂ while the NO₂-SCR reaction only involved NO₂. We calculated the apparent activation energies of NO and NO₂ over Cu_{0.4}-SSZ-13 and Cu_{3.8}-SSZ-13 samples under

FSCR conditions. It can be seen that there is an obvious gap between the E_a of NO and NO₂, indicating separate reaction pathways of NO and NO₂. The E_a of NO₂ was close to the activation barrier under NO₂-SCR conditions. However, the E_a of NO under FSCR conditions was higher than that under SSCR condition (40-80 kJ/mol, Kim et al., *J. Catal.*, **2014**, 311, 447; Kwak et al., *Catal. Lett.*, **2012**, 142, 295.), indicating different reaction pathways of NO under FSCR and SSCR conditions. We calculated the FSCR reaction pathways over various Cu sites as well as BASs (Figs. 4-6 and Figs. S17-19). The results showed that NO tends to react with NH₄NO₃ at BASs under FSCR conditions due to the relatively lower energy barrier of 1.27 eV (122.5 kJ/mol), which is comparable to the experimental values (105-106 kJ/mol, Fig. R1).

Fig. R1 Arrhenius plots of rate versus inverse temperature of NO, NO₂ and NO_x over Cu-SSZ-13 with different Cu loading under FSCR conditions.

19) -On pg 8, the authors write “Cu(II)OH(NH₃)₂”, but the typically described species here is Cu(II)OH (NH₃)₃, which could be the predominant state of Cu(II) depending on the sample composition (10.1021/jacs.6b02651).

Response: Thank you for your valuable comment. We agreed with the reviewer that Cu^{II}OH(NH₃)₃ could be the predominant state in the common situation and modified the FSCR reaction pathway over the NH₃-solvated monomeric Cu-OH site, which has adopted Cu^{II}OH(NH₃)₃ as the initial active site (see Fig. S18). The presence of NO₂ would lead to the formation of fw-Cu^{II} species. Given this situation, in the current version, we have supplied the FSCR pathways over various fw- and NH₃-solvated Cu^{II} [Cu^{II} and (Cu^{II}OH)²⁺] species, BASs and even dimer Cu^I species (**Fig. 4-6 and S17-S19**). The result shows that the overall energy barriers for the FSCR over fixed framework and mobile NH₃-solvated Cu^{II} species (1.54-1.92 eV) are higher than that over the Brønsted acid sites

(1.27 eV). We have revised the DFT calculation section, please see below.

Modification (Lines 254-263 in MS): We first calculated the FSCR reaction pathway over fw-Cu^{II} species (**Fig. 4**). The framework-bound Cu^{II} first adsorbs two NH₃ molecule without separation from the framework, which then interacts with NO₂ to form Z₂Cu^{II}NH₃OH and NH₂NO species (B→C). The Z₂Cu^{II}NH₃OH further adsorbs an NH₃ molecule and reacts with NO, resulting in the formation of Z₂Cu^{II}NH₃, NH₂NO and H₂O (E→F), which was predicted to be the rate-determining step of the SCR reaction cycle with a high energy barrier of 1.92 eV. The formed NH₂NO is easily decomposed into N₂ and H₂O through a series of H-migration and isomerization processes (**Fig. S16**).²⁹ Last, the gaseous NH₃ molecules are supplied to regenerate the initial A species.

(**Lines 276-280 in MS**): In addition, the FSCR reaction pathways over NH₃-solvated Cu^{II} [Cu^{II} and (Cu^{II}OH)²⁺] species were also calculated and presented in **Fig. S17-18**. All the energy barriers were found to be relatively high (1.54 and 1.65 eV). We additionally calculated the FSCR reaction over Cu-dimer sites (**Fig. S19**), which also showed a high energy barrier of 1.58 eV.

Fig. 4 Gibbs free energy profile of the fast SCR cycle at Z₂Cu^{II} site as well as optimized geometries of the reactants, transition states (TSs), intermediates, and products for all elementary steps. Except for the O atoms linked to the Cu²⁺ ion, all other atoms of the zeolite framework are omitted for

clarity. Orange, red, blue, and white circles denote Cu, O, N, and H atoms, respectively.

Fig. 5 Gibbs free energy profile of the fast SCR cycle at $\text{ZCu}^{\text{II}}\text{-OH}$ site as well as optimized geometries of the reactants, transition states (TSs), intermediates, and products for all elementary steps. Except for the two O atoms linked to the Cu-OH group, all other atoms of the zeolite framework are omitted for clarity. All legends are the same as those in **Fig. 4**.

Fig. S17. Gibbs free energy profile of the fast SCR cycle at the NH₃-solvated monomeric Cu^{II} site as well as optimized geometries of the reactants, transition states (TSs), intermediates, and products for all elementary steps. Except for the O atoms linked to the Cu²⁺ ion, all other atoms of the zeolite framework are omitted for clarity. Orange, red, blue, and white circles denote Cu, O, N, and H atoms, respectively.

Fig. S18. Gibbs free energy profile of the fast SCR cycle at the NH_3 -solvated monomeric $\text{Cu}^{\text{II}}\text{OH}$ site as well as optimized geometries of the reactants, TSS, intermediates, and products for all elementary steps. Except for the O atoms linked to the $\text{Cu}^{\text{II}}\text{OH}$ group, all other atoms of the zeolite framework are omitted for clarity. All legends are the same as those in **Fig. S17**.

Fig. S19. Gibbs free energy profile of the fast SCR cycle at the Cu-dimer site as well as optimized geometries of the reactants, transition states, intermediates, and products for all elementary steps. The zeolite framework is omitted for clarity. All legends are the same as those in **Fig. S17**.

• **Typos/ vague language:**

20) -SI, page 5: the authors write “ZHO” but I believe this should be “ZNO”

Response: Thank you for your comment. “ZHO” has been revised into “ZNO”

-Fig S5: please clarify the temperature for each treatment.

Response: Thank you for your comment. The sample was first pretreated in O₂/He at 500 °C for 30 min before decreasing the temperature to 200 °C, after which the Pre. spectra were collected. Then the sample was treated under different atmospheres at 200°C. We have added the temperature in Fig. 3, Fig. S13 and Fig. S14.

21) - Page 7, the authors state “Under FSCR conditions, however, the XANES profile was more like

that of pretreated Cu-SSZ-13, demonstrating that most of the copper species were Cu^+ ions, although there was still a slight amount of Cu^+ ions". I believe the second clause of the sentence should say " Cu^{2+} ions".

Response: Thank you for your comment. It should be Cu^{2+} ions. We have corrected it.

Reviewer #3 (Remarks to the Author):

This is a very well written and presented discussion of a series of measurements and calculations that provide strong support for their conclusions regarding the mechanism of inhibition of SCR activity in Cu-SSZ13 catalysts in the presence of NO₂. The work will be of interest to the growing number of researchers interested in the mechanisms of SCR activity and poisoning of copper zeolite catalysts, and the broader field of environmental catalysis. The combination of careful kinetic analysis, *in situ/operando* XAS studies, and DFT mechanism modeling makes a very compelling case for this conclusion. The results are applicable for the design of regeneration protocols and operational aspects of the active catalysts.

The methods used are appropriate and well described. The arguments are presented clearly and methodically. The paper should be published in Nature Communications, following a few very minor wording fixes and addressing the questions raised below. Line 108 "generally thought to be resulted from.." should be "generally thought to result from..." Line 251 "we found that NO₂ leads to a deep..." (need to add 'to' in this sentence) Line 297 there is a ppm missing after NO in the mixture identification here

Response: Thank you so much for your positive comments. We have revised these sentences. Please see below.

Modification:

Lines 133-135 in MS: The extremely low NO conversion at low temperatures was generally thought to result from zeolite pore blocking by the formation of stable NH₄NO₃.^{21,23}

Lines 315-318 in MS: In summary, by combining analysis of *in situ* spectroscopic measurements with DFT calculations, we found that NO₂ leads to the deep oxidation of copper species as fw-Cu^{II}, which significantly inhibits the locally homogeneous SSCR over dynamic dimer Cu sites.

Lines 360-361 in MS: Then, the sample was exposed to 500 ppm NH₃/He, 500 ppm NO/ He and 500 ppm NH₃/ He + 500 ppm NO/ He for 60 min, respectively.

The experimental preparation of the catalyst resulted in zeolite with an Al/Si ratio of about 5. The XAS and kinetics measurements were carried out on samples with this ratio. In the DFT studies, the model replaced one Si atom in each double six-membered ring with Al, for a Si/Al ratio of 11. The authors should comment on this difference. Is there any reason to expect this difference to affect the conclusions about the effect of NO₂ on the SCR activity?

Response: Thank you for your kind comments. The SSZ-13 model with Si/Al ratio of 11 has been widely adopted in existing DFT studies (*J. Am. Chem. Soc.* **2017**, 139, 4935; *ACS Catal.* **2020**, 10, 5646). Test calculations with a Si/Al ratio of 5 (Fig. R2) showed no significant impact on the energetic profiles (< 0.05 eV, Table R1), indicating that the present models were reasonable for the assessment of the reaction mechanism.

Fig. R2 The SSZ-13 zeolite model used in the DFT calculations. (a) Si/Al=11; (b) Si/Al=5. Yellow, pink, red, and white circles denote Si, Al, O, and H atoms, respectively.

Table R1. Calculated energy barriers of the rate-determining step for the fast SCR cycle over Cu-OH site and BAS in the zeolites with different Si/Al ratio.

Si/Al ratio	Cu-OH site	BAS
11	1.58	1.27
5	1.53	1.24

Otherwise, the paper is interesting, well written, and scientifically sound. It should be of interest to the readers of Nature Communications.

Response: Thank you again for your positive comments.

REVIEWER COMMENTS

Reviewer #1 (Remarks to the Author):

While the authors have satisfactorily dealt with my previous concerns under points 1) – 4), I still partially disagree with their rebuttal to my question 5):

5) What is the evidence for ruling out the formation of NH_3 -ligated, mobile Cu^{2+} species?

Please see below my comments in italics.

Thermodynamic aspects - The DFT results in Fig. S15 emphasize a significant difference between the formation of CuI pairs from $\text{CuI}(\text{OH})(\text{NH}_3)_3$ in comparison to $\text{CuI}(\text{NO}_2)(\text{NH}_3)_3$ and $\text{CuI}(\text{NH}_3)_4$ which should be noted and discussed in more details. In fact, Fig. S15 (a) shows that the dimeric $\text{CuI}(\text{OH})(\text{NH}_3)_3$ is thermodynamically more stable than the isolated configuration (-0.41 eV), contrary to the dimeric complexes from the other two species. The exergonic formation of the same “two proximate” binuclear $\text{CuI}(\text{OH})(\text{NH}_3)_3$ complex of Figure S15 (a) was reported already by Hu et al., *Angew. Chemie* 60 (2021) 7197: this should be mentioned in the revised manuscript.

Kinetic aspects - The high inter-cage diffusional barrier of $\text{CuI}(\text{OH})(\text{NH}_3)_3$ is consistent with MD results from Paolucci et al., *JACS* 138 (2016) 6028, who showed that $\text{CuI}(\text{OH})(\text{NH}_3)_3$ is only intra-cage mobile, while $\text{CuI}(\text{NH}_3)_2$ is compatible with inter-cage diffusion, too. However, CO + NH_3 titration experiments have probed $\text{CuI}(\text{OH})(\text{NH}_3)_x$ dimeric complexes which catalyze the CO oxidation to CO_2 over Cu-SSZ-13 (Villamaina et al., *ChemCatChem*. 12 (2020) 3843). Also, Hu et al., *Angew. Chemie* 60 (2021) 7197 (SI - S10) showed that the one- NH_3 -ligand $\text{CuI}(\text{OH})(\text{NH}_3)$ complex has a very low diffusional barrier of only 12 kJ/mol, and proposed that $\text{CuI}(\text{OH})(\text{NH}_3)$ acts as an inter-cage transportation medium for $\text{CuI}(\text{OH})(\text{NH}_3)_3$. Accordingly, I disagree with the authors' generalized statements “the diffusion of various NH_3 -solvated CuI species into an adjacent cage to form CuI pairs are strongly inhibited due to the extremely high barriers caused by the steric effect “ and “formation of dynamic binuclear sites is strongly inhibited by NO_2 ”: in my view, the formation of dynamic binuclear CuI sites from $\text{CuI}(\text{OH})(\text{NH}_3)_3$ cannot be ruled out on both theoretical and experimental grounds, whether NO_2 is present or not.

On the other hand, I agree with the authors that the formation of CuI pairs from $\text{CuI}(\text{NH}_3)_4$ and $\text{CuI}(\text{NO}_2)(\text{NH}_3)_3$ is hardly possible according to both thermodynamics and kinetics. From an experimental perspective, the unfeasible formation of dimeric $\text{CuI}(\text{NH}_3)_4$ is also consistent with CO + NH_3 titration results (CO oxidation, which is catalyzed by CuI dimers, proceeds only over $\text{CuI}(\text{OH})(\text{NH}_3)_3$, see Villamaina et al., *ChemCatChem*. 12 (2020) 3843) and with the recently proposed hydrolysis mechanism converting $\text{CuI}(\text{NH}_3)_4$ to $\text{CuI}(\text{OH})(\text{NH}_3)_3$, see Hu et al., *ACS Catal.* 11 (2021) 11616). The authors may want to cite these references to further support their theoretical results regarding the unfeasible pairing of $\text{CuI}(\text{NH}_3)_4$.

Reviewer #2 (Remarks to the Author):

While the authors addressed some of my concerns, I still have major concerns about the core interpretations and conclusions of this work, and thus also its impact. Unfortunately I must still recommend rejection on these grounds, and re-submission to a more specialized journal once the technical issues are addressed.

- The central finding of the paper, as stated in the abstract, is as follows:

“strong oxidation by NO₂ forces Cu ions to exist mainly as fixed framework Cu²⁺ species (fw-Cu²⁺), which impede the formation of dynamic binuclear Cu⁺ species that serve as the main active sites for the standard SCR (SSCR) reaction. As a result, the SSCR reaction is significantly inhibited by NO₂ in the zeolite system, and the NO₂-involved SCR reaction occurs with an energy barrier higher than that of the SSCR reaction on dynamic binuclear sites.”

The authors' rebuttal states,

“Although the reference [10.1126/science.aan5630] mentioned that NO₂ accelerates SCR rates by accelerating CuI oxidation kinetics, the sacrifice of the mobility of Cu species due to its high valence and coordination number was not considered, which would significantly affect the SSCR reaction. For the first time, our study reported that NO₂ inhibited the mobility of Cu by oxidizing the Cu species into a high-valence state.”

Elsewhere, the authors state,

“The various CuI species have more difficulty diffusing into the same cage and facilitating O₂ activation to complete the SSCR reaction than CuI(NH₃)₂ species, due to the strong interaction with the zeolite framework and steric effects (see Fig. S15). As a result, the SSCR reaction with the participation of dimeric Cu was significantly inhibited.”

Unfortunately, I believe the authors' argument is logically flawed.

The literature agrees that SSCR occurs via the following redox cycle:

NH₃-solvated Cu(I) is oxidized to binuclear Cu(II) sites by O₂ ; NH₃-solvated Cu(II) is reduced to Cu(I) by NO + NH₃

The literature has also shown that NO₂ oxidizes Cu(I), consistent with the present manuscript. The authors argue that NO₂-inhibition of SCR rates arises from NO₂ forming un-reactive, framework-bound Cu(II) sites, which inhibit the formation of dynamic multinuclear species, thus inhibiting SSCR.

However, a preponderance of literature reports has shown that initially framework-bound Cu(II) sites can be reduced by NO + NH₃ (using XAS and titration techniques):

Refs: <https://doi.org/10.1021/acscatal.0c05362> , <https://doi.org/10.1002/anie.202014926>,
<https://doi.org/10.1002/anie.201407030>, <https://doi.org/10.1016/j.jcat.2020.05.022>.,
<https://doi.org/10.1021/cs501673g>.

The authors' argument that oxidation by NO₂ inhibits the formation of dynamic binuclear sites for SSCr does not explain the inhibition of NO conversion. Fast SCR consumes both NO and NO₂; the proposal that oxidation of Cu(I) by NO₂ inhibits NO conversion by forming framework-bound Cu(II) does not make sense, given that framework Cu(II) is readily reduced in NO and NH₃ (consuming NO).

The authors' claim that "Cu(I) species have more difficulty diffusing into the same cage and facilitating O₂ activation to complete the SSCr reaction" does not make sense; O₂ activation occurs on Cu(I) sites, not Cu(II) sites.

The only way the authors' argument might make sense if they were to claim that NO₂-oxidation of Cu(I) forms a distinct and highly un-reactive NO₂-Cu(II) complex, not just a regular framework-bound Cu(II) species. Additional evidence would be needed to support such a claim.

- Furthermore, couldn't the NO₂-inhibition effects in this paper be more easily rationalized by NH₄NO₃ accumulation that blocks active sites? How can the authors rule out this explanation as the dominant reason for inhibition?

- Concerning XAS results:

I remain concerned about the authors' XANES results for their Cu-CHA catalyst following reduction by NO + NH₃ at 200°C. The low pre-edge feature suggests a large fraction of irreducible Cu(II) sites, implying that these sites are not extraframework Cu cations, which would invalidate some of the arguments in this paper.

- Concerning control experiments with H-form zeolites:

Thank you for adding these experiments. In Fig S9, I cannot tell whether the catalyst mass was kept constant. Assuming this is the case, why did Si/Al 6, 12, and 18 have fairly similar NO_x conversions (30-40%) when the acid site density should be varying by more than 2x?

- The authors claim that "there have been few studies reporting that NO₂ measurably promotes the NH₃-SCR efficiency over Cu-SSZ-13 catalytic systems."

I agree with the authors that their results clearly show an NO₂-inhibition effect. However, they should comment on the multiple literature reports showing that fast SCR rates (with NO + NO₂) are higher than standard SCR (with NO + O₂):

<https://doi.org/10.1021/acs.jpcc.1c06651>, <https://doi.org/10.4271/2007-01-1575>.

Reviewer #3 (Remarks to the Author):

The authors have certainly responded appropriately to the minor concerns I had about this paper. In addition, it appears that they have responded appropriately, with modifications in some instances and with further explanations in others, to the useful comments and suggestions of the other two reviewers. I support publishing this revised manuscript in Nature Communications.

Point-by-Point Response to the Reviewers' Comments

Reviewer #1 (Remarks to the Author):

While the authors have satisfactorily dealt with my previous concerns under points 1) – 4), I still partially disagree with their rebuttal to my question 5):

5) What is the evidence for ruling out the formation of NH_3 -ligated, mobile Cu^{2+} species? Please see below my comments in italics.

Thermodynamic aspects - *The DFT results in Fig. S15 emphasize a significant difference between the formation of Cu^{II} pairs from $\text{Cu}^{\text{II}}\text{OH}(\text{NH}_3)_3$ in comparison to $\text{Cu}^{\text{II}}\text{NO}_2(\text{NH}_3)_3$ and $\text{Cu}^{\text{II}}(\text{NH}_3)_4$ which should be noted and discussed in more details. In fact, Fig. S15 (a) shows that the dimeric $\text{Cu}^{\text{II}}\text{OH}(\text{NH}_3)_3$ is thermodynamically more stable than the isolated configuration (-0.41 eV), contrary to the dimeric complexes from the other two species. The exergonic formation of the same “two proximate” binuclear $\text{Cu}^{\text{II}}\text{OH}(\text{NH}_3)_3$ complex of Figure S15 (a) was reported already by Hu et al., *Angew. Chemie* 60 (2021) 7197: this should be mentioned in the revised manuscript.*

Kinetic aspects - *The high inter-cage diffusional barrier of $\text{Cu}^{\text{II}}\text{OH}(\text{NH}_3)_3$ is consistent with MD results from Paolucci et al., *JACS* 138 (2016) 6028, who showed that $\text{Cu}^{\text{II}}\text{OH}(\text{NH}_3)_3$ is only intra-cage mobile, while $\text{Cu}^{\text{I}}(\text{NH}_3)_2$ is compatible with inter-cage diffusion, too. However, CO + NH_3 titration experiments have probed $\text{Cu}^{\text{II}}\text{OH}(\text{NH}_3)_x$ dimeric complexes which catalyze the CO oxidation to CO_2 over Cu-SSZ-13 (Villamaina et al., *ChemCatChem*. 12 (2020) 3843). Also, Hu et al., *Angew. Chemie* 60 (2021) 7197 (SI - S10) showed that the one- NH_3 -ligand $\text{Cu}^{\text{II}}\text{OH}(\text{NH}_3)$ complex has a very low diffusional barrier of only 12 kJ/mol, and proposed that $\text{Cu}^{\text{II}}\text{OH}(\text{NH}_3)$ acts as an inter-cage transportation medium for $\text{Cu}^{\text{II}}\text{OH}(\text{NH}_3)_3$. Accordingly, I disagree with the authors' generalized statements “the diffusion of various NH_3 -solvated Cu^{II} species into an adjacent cage to form Cu^{II} pairs are strongly inhibited due to the extremely high barriers caused by the steric effect “ and “formation of dynamic binuclear sites is strongly inhibited by NO_2 ”: in my view, the formation of dynamic binuclear Cu^{II} sites from $\text{Cu}^{\text{II}}\text{OH}(\text{NH}_3)_3$ cannot be ruled out on both theoretical and experimental grounds, whether NO_2 is present or not.*

*On the other hand, I agree with the authors that the formation of Cu^{II} pairs from $\text{Cu}^{\text{II}}(\text{NH}_3)_4$ and $\text{Cu}^{\text{II}}\text{NO}_2(\text{NH}_3)_3$ is hardly possible according to both thermodynamics and kinetics. From an experimental perspective, the unfeasible formation of dimeric $\text{Cu}^{\text{II}}(\text{NH}_3)_4$ is also consistent with CO + NH_3 titration results (CO oxidation, which is catalyzed by Cu^{II} dimers, proceeds only over $\text{Cu}^{\text{II}}\text{OH}(\text{NH}_3)_3$, see Villamaina et al., *ChemCatChem*. 12 (2020) 3843) and with the recently proposed hydrolysis mechanism converting $\text{Cu}^{\text{II}}(\text{NH}_3)_4$ to $\text{Cu}^{\text{II}}\text{OH}(\text{NH}_3)_3$, see Hu et al., *ACS Catal.* 11 (2021) 11616). The authors may want to cite these references to further support their theoretical results regarding the unfeasible pairing of $\text{Cu}^{\text{II}}(\text{NH}_3)_4$.*

Response: Thank you so much for your valuable comments. We agree with the reviewer that the formation of Cu^{II} dimers from Cu^{II}OH(NH₃)₃ cannot be ruled out. We have cited these related references to improve our manuscript. The relevant modification is as follows:

Modification (Lines 260-262 in MS): Therefore, we next turned to the DFT calculation to investigate the possible FSCR reaction pathways over fw- and NH₃-solvated Cu^{II} [Cu²⁺ and (Cu^{II}OH)⁺] species, BASs and dimer Cu^I species.

(Lines 287-301 in MS): Moreover, we consider the possibility that various NH₃-solvated Cu^{II} species diffuse into an adjacent cage to form Cu^{II}-pairs as shown in **Fig. S15**. As expected, the formation of Cu^{II} pairs from Cu^{II}(NH₃)₄ and Cu^{II}NO₂(NH₃)₃ is both thermodynamically and kinetically inhibited due to the steric effect as well as strong interaction with zeolite framework. However, it should be noted that Cu^{II}OH(NH₃)₃ is different from the other forms since binuclear Cu^{II}OH(NH₃)₃ in one cage is thermodynamically more stable than the isolated configuration. Villamaina et al. validated the formation of Cu^{II}(OH)(NH₃)_x dimeric complexes in the oxidation atmosphere through CO + NH₃ titration experiment.⁴¹ Hu et al. proposed that Cu^{II}(OH)(NH₃), which is structurally similar to Cu^I(NH₃)₂ that has one positive charge and two ligands, acts as inter-cage transportation medium.⁶ The Z2Cu^{II} species can transform ZCu^{II}(OH) by NH₃-assisted hydrolysis to achieve the Cu pairing.⁴² However, the regeneration of Cu^{II}(OH) in dimeric form showed a high energy barrier of 1.58 eV (A→B in **Fig. S19**), suggesting that the dimeric Cu^{II}(OH) species were not highly active in the FSCR reaction.

Reviewer #2 (Remarks to the Author):

While the authors addressed some of my concerns, I still have major concerns about the core interpretations and conclusions of this work, and thus also its impact. Unfortunately I must still recommend rejection on these grounds, and re-submission to a more specialized journal once the technical issues are addressed.

• The central finding of the paper, as stated in the abstract, is as follows:

“strong oxidation by NO₂ forces Cu ions to exist mainly as fixed framework Cu²⁺ species (fw-Cu²⁺), which impede the formation of dynamic binuclear Cu⁺ species that serve as the main active sites for the standard SCR (SSCR) reaction. As a result, the SSCR reaction is significantly inhibited by NO₂ in the zeolite system, and the NO₂-involved SCR reaction occurs with an energy barrier higher than that of the SSCR reaction on dynamic binuclear sites.”

The authors' rebuttal states,

“Although the reference [10.1126/science.aan5630] mentioned that NO₂ accelerates SCR rates by accelerating Cu^I oxidation kinetics, the sacrifice of the mobility of Cu species due to its high valence and coordination number was not considered, which would significantly affect the SSCR reaction. For the first time, our study reported that NO₂ inhibited the mobility of Cu by oxidizing the Cu species into a high-valence state.”

Elsewhere, the authors state,

“The various Cu^{II} species have more difficulty diffusing into the same cage and facilitating O₂ activation to complete the SSCR reaction than Cu^I(NH₃)₂ species, due to the strong interaction with the zeolite framework and steric effects (see Fig. S15). As a result, the SSCR reaction with the participation of dimeric Cu was significantly inhibited.”

Unfortunately, I believe the authors' argument is logically flawed.

The literature agrees that SSCR occurs via the following redox cycle:

NH₃-solvated Cu(I) is oxidized to binuclear Cu(II) sites by O₂; NH₃-solvated Cu(II) is reduced to Cu(I) by NO + NH₃

The literature has also shown that NO₂ oxidizes Cu(I), consistent with the present manuscript. The authors argue that NO₂-inhibition of SCR rates arises from NO₂ forming un-reactive, framework-bound Cu(II) sites, which inhibit the formation of dynamic multinuclear species, thus inhibiting SSCR.

However, a preponderance of literature reports has shown that initially framework-bound Cu(II) sites can be reduced by NO + NH₃ (using XAS and titration techniques):

Refs: <https://doi.org/10.1021/acscatal.0c05362>,
<https://doi.org/10.1002/anie.202014926>, <https://doi.org/10.1002/anie.201407030>,
<https://doi.org/10.1016/j.jcat.2020.05.022>, <https://doi.org/10.1021/cs501673g>.

Response: Thank you so much for your valuable comments. We agreed with the

reviewer that framework-bound Cu(II) sites can be reduced by NO + NH₃ (even in the presence of O₂ as SSCR conditions). However, the situation is different in the presence of NO₂. As shown in **Fig. 3h** and **Fig. S13c**, most of Cu^{II} species are not reduced under FSCR conditions (i.e., the co-existence of NO, NH₃ and NO₂). All the literatures referred by the reviewer showed the reduction of Cu^{II} by NO + NH₃ in the absence of NO₂.

The authors' argument that oxidation by NO₂ inhibits the formation of dynamic binuclear sites for SSCR does not explain the inhibition of NO conversion. Fast SCR consumes both NO and NO₂; the proposal that oxidation of Cu(I) by NO₂ inhibits NO conversion by forming framework-bound Cu(II) does not make sense, given that framework Cu(II) is readily reduced in NO and NH₃ (consuming NO).

Response: Thank you for your valuable comments. We revised the related sentences to express accurately. The strong oxidation of NO₂ forces Cu ions to exist mainly as Cu^{II} species (fw-Cu²⁺ and NH₃-solvated Cu^{II} with high CNs), which impede the mobility of Cu species. The SCR reaction (involved NO or/and NO₂) occurred at the Cu^{II} sites showed a higher energy barrier than that of the standard SCR reaction on dynamic binuclear sites. In addition, we observed that the framework-bound Cu^{II} is not readily reduced by NO and NH₃ **in the presence of NO₂** as shown in Fig.3h and Fig. S13c. Moreover, as shown in **Fig. S18**, we found that the Cu^{II}OH is reduced by NO + NH₃ to form Cu^I(NH₃)₂ intermediate, which is instantly oxidized by NO₂ without any barriers (B→D in Fig. S18). The subsequent reaction from the formed Cu^{II}NO₂(NH₃)₃ with a high energy barrier of 1.65 eV is the rate-determining step of this FSCR cycle, resulting in an accumulation of the unreactive Cu^{II} species.

Modification (Lines 22-26 in MS): strong oxidation by NO₂ forces Cu ions to exist mainly as Cu^{II} species (fw-Cu^{II} and NH₃-solvated Cu^{II} with high CNs), which impede the mobility of Cu species. The SCR reaction occurred at these Cu^{II} sites with weak mobility showed a higher energy barrier than that of the standard SCR reaction on dynamic binuclear sites.

The authors' claim that "Cu^{II} species have more difficulty diffusing into the same cage and facilitating O₂ activation to complete the SSCR reaction" does not make sense; O₂ activation occurs on Cu(I) sites, not Cu(II) sites.

The only way the authors' argument might make sense if they were to claim that NO₂-oxidation of Cu(I) forms a distinct and highly un-reactive NO₂-Cu(II) complex, not just a regular framework-bound Cu(II) species. Additional evidence would be needed to support such a claim.

Response: Thank you for your valuable comments. We apologize that the expression is unclear and hence delete it. We agree with the reviewer that the O₂ activation occurs

on Cu^I sites rather than Cu^{II} sites. The presence of NO₂ leads to the domination of Cu^{II} species (fw-Cu^{II} and NH₃-solvated Cu^{II} with high CNs). The fw-Cu^{II} species were validated by the observation of second-shell scattering peak in Fig. 3h. The NH₃-solvated Cu^{II} species were validated by the observation of NH₃ desorption from Cu sites in FSCR-TPD profiles (Fig. S5). The energy barriers of the NO₂-involved SCR reactions over these Cu^{II} species are higher than that of the SSCR reaction on dynamic binuclear sites (Figs. 4-5 and Figs. S17-18). It is worth mentioning that the Cu^{II}NO₂(NH₃)₃ appeared in the DFT-calculated FSCR cycles over the NH₃-solvated Cu^{II} and Cu^{II}OH sites (A in Fig. S17 and D in Fig. S18). However, the high energy barriers indicate the low reactivity of the NO₂-Cu^{II} complex as the reviewer mentioned.

- Furthermore, couldn't the NO₂-inhibition effects in this paper be more easily rationalized by NH₄NO₃ accumulation that blocks active sites? How can the authors rule out this explanation as the dominant reason for inhibition?

Response: Thank you for your valuable comments. The accumulation of NH₄NO₃, which suppresses the mass transfer process during SCR reaction, is also an important reason for the NO₂-inhibition effects indeed. However, the block of active sites by NH₄NO₃ is not the only reason since NO₂ conversion still proceeded while NO conversion was totally inhibited regardless of Cu loading as shown in **Fig. S2** at low temperatures under FSCR conditions. This phenomenon was also observed in previous studies (*J. Phys. Chem. C*, 2018, 122, 25948; *Emiss. Control. Sci. Technol.*, 2019, 5, 124) that NO conversion was significantly lower than NO₂ conversion under FSCR conditions. In this paper, we first reported that the presence of NO₂ significantly changes the state and coordination of active Cu sites and the dominant mononuclear Cu^{II} species lead a high energy barrier of NO reduction, thereby revealing a previously unreported chemical mechanism for the NO₂-inhibition effects.

Modification (Lines 133-141 in MS): The extremely low NO conversion at low temperatures was previously thought to be resulted from zeolite pore blocking by the formation of stable NH₄NO₃.^{21,23} The NH₄NO₃ formation was verified by the observation of N₂O in an FSCR-TPD experiment (**Fig. S5**), since the N₂O mainly originated from NH₄NO₃ decomposition. Interestingly, the NO₂ reduction markedly decreased with the increase in Cu loading, while it increased as the number of BASs rose at low temperatures (Figs. 2b-2c and Fig. S6). This demonstrated that the block of active sites by NH₄NO₃ was not the only reason for the NO₂-inhibition effects, otherwise both NO and NO₂ reduction were inhibited.

• **Concerning XAS results:**

I remain concerned about the authors' XANES results for their Cu-CHA catalyst

following reduction by NO + NH₃ at 200°C. The low pre-edge feature suggests a large fraction of irreducible Cu(II) sites, implying that these sites are not extraframework Cu cations, which would invalidate some of the arguments in this paper.

Response: Thank you for your valuable comments. We agree with that there are probably some irreducible Cu-oxides in our Cu3.8-SSZ-13 sample due to its relatively high Cu loadings. However, it can be noted that CuO_x species were not obviously detected by our *ex-situ* XANES and EXAFS results (**Fig. S12**), indicating the amount of CuO_x species is insignificant. Moreover, it can be expected that the existence of CuO_x in Cu3.8-SSZ-13 sample has no significant influence on the conclusions of this paper since the kinetic studies were conducted with gradient Cu loading and the difference of active Cu^{II}/Cu^I distribution under SSCR and FSCR conditions is independent of CuO_x. The primary conclusion of XAFS is that NO₂ has stronger oxidation than O₂, leading to the domination of Cu^{II} species (fw-Cu^{II} and NH₃-solvated Cu^{II}). The results showed that relatively more Cu^I species existed under SSCR conditions while FSCR conditions favored the presence of dominated Cu^{II} species, which is irrelevant to CuO_x. Similar XAFS results were also reported by previous studies (McEwen et al., *Catal. Today*, 2012, 184, 129; Paolucci et al., *Science*, 2017, 357, 898).

Modification (Lines 131-135 in SI): However, it should be noted that the feature B is not as high as reported in literatures^{17,21}, indicating that there were probably some irreducible Cu-oxides in the Cu3.8-SSZ-13 sample. Nevertheless, the amount of CuO_x was expected to be insignificant since no obvious CuO_x species were detected in *ex-situ* XAFS by comparing with CuO_x standards.

• **Concerning control experiments with H-form zeolites:**

Thank you for adding these experiments. In Fig S9, I cannot tell whether the catalyst mass was kept constant. Assuming this is the case, why did Si/Al 6, 12, and 18 have fairly similar NO_x conversions (30-40%) when the acid site density should be varying by more than 2x?

Response: Thank you for your valuable comments. The mass of the three samples was kept constant (~60 mg). We reevaluated the NO₂-SCR activity over the three H-zeolites and the same results were obtained. The results of Fig. S9 showed that NO₂ conversion increased with the decrease of Si/Al, indicating that NO₂ primarily reacted at BASs due to abundant BASs of H-SSZ-13 with low Si/Al. However, as you stated, the difference of NO_x conversion was inapparent, which was probably attributed to the NH₄NO₃ formation. As previous studies, the BASs were responsible for the formation of NH₄NO₃ (Liu et al., *J Phys. Chem. C*, 2021, 125, 25; Shan et al., *Appl. Catal. B*, 2020, 275, 119105), resulting in more NH₄NO₃ formation for H-SSZ-13 with lower Si/Al. Therefore, although the low-Si/Al H-SSZ-13 possesses more BASs that favors the NO₂-SCR reaction, the formation of more NH₄NO₃ blocks the active

sites.

- The authors claim that “there have been few studies reporting that NO₂ measurably promotes the NH₃-SCR efficiency over Cu-SSZ-13 catalytic systems.”

I agree with the authors that their results clearly show an NO₂-inhibition effect. However, they should comment on the multiple literature reports showing that fast SCR rates (with NO + NO₂) are higher than standard SCR (with NO + O₂):

<https://doi.org/10.1021/acs.jpcc.1c06651>, <https://doi.org/10.4271/2007-01-1575>.

Response: Thank you for your kind suggestions. We have commented on and cited the literatures you referred as well as other literatures (Shan et al., *J. Phys. Chem. C*, 2018, 122, 25948, Kwak et al., *Catal. Lett.*, 2012, 142, 295) that showed higher fast SCR rates than standard SCR rates. In fact, NO₂ primarily inhibits NO conversion rather than NO_x conversion. We added the rates of NO and NO₂ over Cu_{0.4}-SSZ-13 sample under different reaction conditions as shown in Fig. S11b. It can be seen that the NO conversion was inhibited under FSCR conditions compared to that under SSCR conditions while NO₂ showed relatively high conversion, resulting in a higher NO_x conversion than that under SSCR conditions. Therefore, the higher NO_x conversion under FSCR conditions than that under SSCR conditions reported in the literatures were primarily resulted from high NO₂ conversion while NO conversion was still inhibited (Bendrich et al., *Appl. Catal. B*, 2018, 222, 76; Cui et al., *Emiss. Control. Sci. Technol.*, 2019, 5, 124; Shan et al., *J. Phys. Chem. C*, 2018, 122, 25948).

Modification (Lines 161-171 in MS): For Cu_{0.4}-SSZ-13 sample, the NO conversion under FSCR conditions was likewise inhibited compared to that under SSCR conditions. Differently, the NO₂ conversion under FSCR and NO₂-SCR conditions were relatively higher than the NO conversion under SSCR conditions due to the insufficient Cu active sites for SSCR reaction. As a result, the FSCR rates of NO_x can also be higher than SSCR rates of NO_x especially when the Cu-zeolite behaves low NO conversion (low Cu loadings, hydrothermal aging state, etc.), which was observed in previous studies.^{23,27,36,37} In another word, the NO conversion was inhibited in the presence of NO₂, while the effect of NO₂ on NO_x conversion was uncertain and relates to NO₂ conversion under FSCR conditions as well as NO_x conversion under SSCR conditions.

Fig. S11 NO and NO_2 reaction rates over (a) $\text{Cu}_{2.6}$ -SSZ-13 and (b) $\text{Cu}_{0.4}$ -SSZ-13 catalysts under SSCR, FSCR and NO_2 -SCR conditions.

Reviewer #3 (Remarks to the Author): The authors have certainly responded appropriately to the minor concerns I had about this paper. In addition, it appears that they have responded appropriately, with modifications in some instances and with further explanations in others, to the useful comments and suggestions of the other two reviewers. I support publishing this revised manuscript in Nature Communications.

Response: Thank you so much again for your positive comments.

REVIEWERS' COMMENTS

Reviewer #1 (Remarks to the Author):

The authors have properly addressed my concerns. I think the paper is now suitable for publication.